# Effects of noise and metabolic cost on cortical task representations

Jake Patrick Stroud[1]\*, Michal Wojcik[2], Kristopher Torp Jensen[1], Makoto Kusunoki[2], Mikiko Kadohisa[2], Mark J Buckley[2], John Duncan[3], Mark G Stokes[2,4], Mate Lengyel[1,5]

[1]Computational and Biological Learning Lab, Department of Engineering, University of Cambridge, Cambridge, United Kingdom; [2]Department of Experimental Psychology, University of Oxford, Oxford, United Kingdom; [3]MRC Cognition and Brain Sciences Unit, University of Cambridge, Cambridge, United Kingdom; [4]Oxford Centre for Human Brain Activity, Wellcome Centre for Integrative Neuroimaging, Department of Psychiatry, University of Oxford, Oxford, United Kingdom; [5]Center for Cognitive Computation, Department of Cognitive Science, Central European University, Budapest, Hungary

## eLife assessment

This work provides a **valuable** analysis of the effect of two commonly used hyperparameters, noise amplitude and firing rate regularization, on the representations of relevant and irrelevant stimuli in trained recurrent neural networks (RNNs). The results suggest an interesting interpretation of prefrontal cortex (PFC) dynamics, based on comparisons to previously published data from the same lab, in terms of decreasing metabolic cost during learning. The evidence indicating that the mechanisms identified in the RNNs are the same ones operating in PFC was considered **incomplete**, but could potentially be bolstered by additional analyses and appropriate revisions.

\*For correspondence:
j.stroud@eng.cam.ac.uk

**Competing interest:** The authors declare that no competing interests exist.

**Abstract** Cognitive flexibility requires both the encoding of task-relevant and the ignoring of task-irrelevant stimuli. While the neural coding of task-relevant stimuli is increasingly well understood, the mechanisms for ignoring task-irrelevant stimuli remain poorly understood. Here, we study how task performance and biological constraints jointly determine the coding of relevant and irrelevant stimuli in neural circuits. Using mathematical analyses and task-optimized recurrent neural networks, we show that neural circuits can exhibit a range of representational geometries depending on the strength of neural noise and metabolic cost. By comparing these results with recordings from primate prefrontal cortex (PFC) over the course of learning, we show that neural activity in PFC changes in line with a minimal representational strategy. Specifically, our analyses reveal that the suppression of dynamically irrelevant stimuli is achieved by activity-silent, sub-threshold dynamics. Our results provide a normative explanation as to why PFC implements an adaptive, minimal representational strategy.

## Introduction

How systems solve complex cognitive tasks is a fundamental question in neuroscience and artificial intelligence (*Rigotti et al., 2013*; *Yang et al., 2019*; *Wang et al., 2018*; *Silver et al., 2016*; *Jensen et al., 2023*). A key aspect of complex tasks is that they often involve multiple types of stimuli, some of which can even be *irrelevant* for performing the correct behavioral response (*Freedman et al., 2001*; *Mante et al., 2013*; *Parthasarathy et al., 2017*) or predicting reward (*Bernardi et al., 2020*;

*Chadwick et al., 2023*). Over the course of task exposure, subjects must typically identify which stimuli are relevant and which are irrelevant. Examples of irrelevant stimuli include those that are irrelevant at all times in a task (*Freedman et al., 2001*; *Chadwick et al., 2023*; *Duncan, 2001*; *Rainer et al., 1998*; *Stokes et al., 2013*) – which we refer to as *static* irrelevance – and stimuli that are relevant at some time points but are irrelevant at other times in a trial – which we refer to as *dynamic* irrelevance (e.g. as is often the case for context-dependent decision-making tasks *Mante et al., 2013*; *Flesch et al., 2022*; *Monsell, 2003*; *Braver, 2012*). Although tasks involving irrelevant stimuli have been widely used, it remains an open question as to how different types of irrelevant stimuli should be represented, in combination with relevant stimuli, to enable optimal task performance.

One may naively think that statically irrelevant stimuli should always be suppressed. However, stimuli that are currently irrelevant may be relevant in a future task. Furthermore, it is unclear whether dynamically irrelevant stimuli should be suppressed at all since the information is ultimately needed by the circuit. It may therefore be beneficial for a neural circuit to represent irrelevant information as long as no unnecessary costs are incurred and task performance remains high. Several factors could have a strong impact on whether irrelevant stimuli affect task performance. For example, levels of neural noise in the circuit as well as energy constraints and the metabolic costs of overall neural activity (*Flesch et al., 2022*; *Whittington et al., 2022*; *Sussillo et al., 2015*; *Orhan and Ma, 2019*; *Löwe, 2023*; *Cueva and Wei, 2018*; *Luo et al., 2023*; *Kao et al., 2021*; *Deneve et al., 2001*; *Barak et al., 2013*) can affect how stimuli are represented in a neural circuit. Indeed, both noise and metabolic costs are factors that biological circuits must contend with (*Tomko and Crapper, 1974*; *Laughlin, 2001*; *Churchland et al., 2006*; *Hasenstaub et al., 2010*). Despite these considerations, models of neural population codes, including hand-crafted models and optimized artificial neural networks, typically use only a very limited range of the values of such factors (*Wang et al., 2018*; *Mante et al., 2013*; *Sussillo et al., 2015*; *Barak et al., 2013*; *Cueva et al., 2020*; *Driscoll et al., 2022*; *Song et al., 2016*; *Echeveste et al., 2020*; *Stroud et al., 2021*) (but also see *Yang et al., 2019*; *Orhan and Ma, 2019*). Therefore, despite the success of recent comparisons between neural network models and experimental recordings (*Wang et al., 2018*; *Mante et al., 2013*; *Sussillo et al., 2015*; *Cueva and Wei, 2018*; *Barak et al., 2013*; *Cueva et al., 2020*; *Echeveste et al., 2020*; *Stroud et al., 2021*; *Lindsay, 2021*), we may only be recovering very few out of a potentially large range of different representational strategies that neural networks could exhibit (*Schaeffer et al., 2022*).

One challenge for distinguishing between different representational strategies, particularly when analyzing experimental recordings, is that some stimuli may simply be represented more strongly than others. In particular, we might expect stimuli to be strongly represented in cortex a priori if they have previously been important to the animal. Indeed, being able to represent a given stimulus when learning a new task is likely a prerequisite for learning whether it is relevant or irrelevant in that particular context (*Rigotti et al., 2013*; *Bernardi et al., 2020*). Previously, it has been difficult to distinguish between whether a given representation existed a priori or emerged as a consequence of learning because neural activity is typically only recorded *after* a task has already been learned. A more relevant question is how the representation *changes* over learning (*Chadwick et al., 2023*; *Reinert et al., 2021*; *Durstewitz et al., 2010*; *Schuessler et al., 2020*; *Costa et al., 2017*), which provides insights into how the specific task of interest affects the representational strategy used by an animal or artificial network (*Richards et al., 2019*).

To resolve these questions, we optimized recurrent neural networks on a task that involved two types of irrelevant stimuli. One feature of the stimulus was statically irrelevant, and another feature of the stimulus was dynamically irrelevant. We found that, depending on the neural noise level and metabolic cost that was imposed on the networks during training, a range of representational strategies emerged in the optimized networks, from maximal (representing all stimuli) to minimal (representing only relevant stimuli). We then compared the strategies of our optimized networks with *learning-resolved* recordings from the prefrontal cortex (PFC) of monkeys exposed to the same task. We found that the representational geometry of the neural recordings changed in line with the minimal strategy. Using a simplified model, we derived mathematically how the strength of relevant and irrelevant coding depends on the noise level and metabolic cost. We then confirmed our theoretical predictions in both our task-optimized networks and neural recordings. By reverse-engineering our task-optimized networks, we also found that activity-silent, sub-threshold dynamics led to the suppression of dynamically irrelevant stimuli, and we confirmed predictions of this mechanism in our neural recordings.

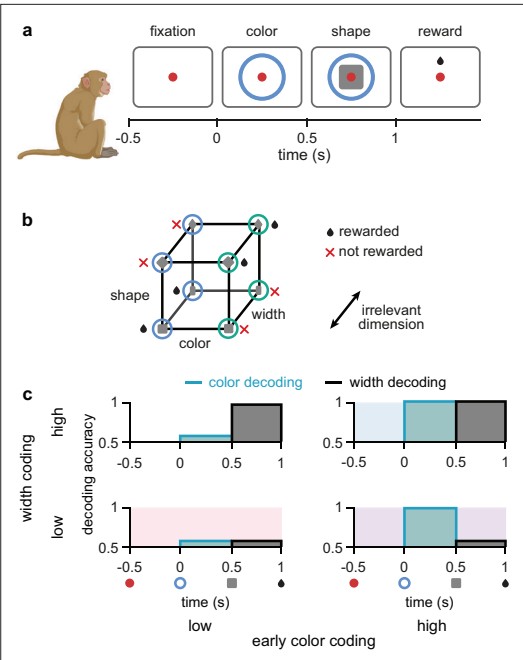

**Figure 1.** Task design and irrelevant stimulus representations. (**a**) Illustration of the timeline of task events in a trial with the corresponding displays and names of task periods. Red dot shows fixation ring, blue (or green) circles appear during the color and shape periods, gray squares (or diamonds) appear during the shape period, and a juice reward is given during the reward period for rewarded combinations of color and shape stimuli (see panel b). No behavioral response was required for the monkeys as it was a passive object–association task (*Wójcik, 2023*). (**b**) Schematic showing that rewarded conditions of color and shape stimuli follows an XOR structure. In addition, the width of the shape was not predictive of reward and was thus an irrelevant stimulus dimension. (**c**) Schematic of four possible representational strategies, as indicated by linear decoding of population activity, for the task shown in panels a and b. Turquoise lines with shading show early color decoding and black lines with shading show width decoding. Strategies are split according to whether early color decoding is low (left column) or high (right column), and whether width decoding is low (bottom row) or high (top row).

In summary, we provide a mechanistic understanding of how different representational strategies can emerge in both biological and artificial neural circuits over the course of learning in response to salient biological factors such as noise and metabolic costs. These results in turn explain why PFC appears to employ a minimal representational strategy by filtering out task-irrelevant information.

## Results

### A task involving relevant and irrelevant stimuli

We study a task used in previous experimental work that uses a combination of multiple, relevant and irrelevant stimuli (*Wójcik, 2023*; *Figure 1a*). The task consists of an initial 'fixation' period, followed by a 'color' period, in which one of two colors are presented. After this, in the 'shape' period, either a square or diamond shape is presented (while the color stimulus stays on), such that the width of the shape can be either thick or thin. After this, the stimuli disappear and reward is delivered according to an XOR structure between color and shape (*Figure 1b*). Note that the width of the shape is not predictive of reward, and it is therefore an *irrelevant* stimulus dimension (*Figure 1b*). As width information is always irrelevant when it is shown, its irrelevance is *static*. In contrast, color is relevant during the shape period but could be ignored during the color period without loss of performance. Hence its irrelevance is *dynamic*.

Due to the existence of multiple different forms of irrelevant stimuli, there exist multiple different representational strategies for a neural circuit solving the task in *Figure 1a*. These representational strategies can be characterized by assessing the extent to which different stimuli are linearly decodable from neural population activity (*Bernardi et al., 2020*; *Stokes et al., 2013*; *Meyers et al., 2008*; *King and Dehaene, 2014*). We use linear decodability because it only requires the computation of simple weighted sums of neural responses, and as such, it is a widely accepted criterion for the usefulness of a neural representation (*DiCarlo and Cox, 2007*). Moreover, while representational strategies can differ along several dimensions in this task (e.g. the decodability of color or shape during the shape period – both of which are task-relevant *Wójcik, 2023*), our main focus here is on the two dimensions that specifically control the representation of irrelevant stimuli. Therefore, depending on whether each of the irrelevant stimuli are linearly decodable during their respective period of irrelevance, we distinguish four different (extreme) strategies, ranging from a 'minimal strategy', in which the irrelevant stimuli are only weakly represented (*Figure 1c*, bottom left; pink shading), to a 'maximal strategy', in which both irrelevant stimuli are strongly represented (*Figure 1c*, top right; blue shading).

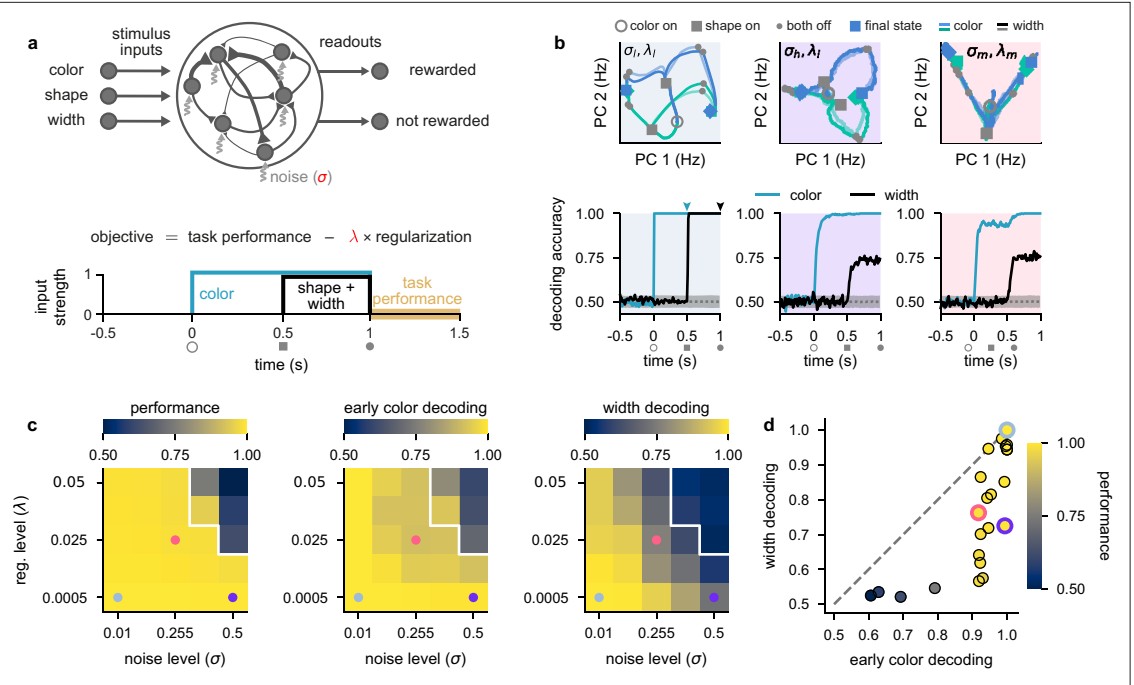

**Figure 2.** Stronger levels of noise and firing rate regularization lead to suppression of task-irrelevant stimuli in optimized recurrent networks. (**a**) Top: illustration of a recurrent neural network model where each neuron receives independent white noise input with strength $\sigma$ (middle). Color, shape, and width inputs are delivered to the network via three input channels (left). Firing rate activity is read out into two readout channels (either rewarded or not rewarded; right). All recurrent weights in the network, as well as weights associated with the input and readout channels, were optimized (Neural network models). Bottom: cost function used for training the recurrent neural networks (*Equation 4*) and timeline of task events within a trial for the recurrent neural networks (*Figure 1a*). Yellow shading on time axis shows the time period in which the task performance term enters the cost function (*Equation 4*). (**b**) Top: neural firing rate trajectories in the top two PCs for an example network over the course of a trial (from color stimulus onset) for a particular noise ($\sigma$) and regularization ($\lambda$) regime. Open gray circles indicate color onset, filled gray squares indicate shape onset, filled gray circles indicate offset of both stimuli, and colored thick and thin squares and diamonds indicate the end of the trial at 1.5 s for all stimulus conditions. Pale and brightly colored trajectories indicate the two width conditions. We show results for networks exposed to low noise and low regularization ($\sigma_l, \lambda_l$; left, pale blue shading), high noise and low regularization ($\sigma_h, \lambda_l$; middle, purple shading), and medium noise and medium regularization ($\sigma_m, \lambda_m$; right, pink shading). Bottom: performance of a linear decoder (mean over 10 networks) trained at each time point within the trial to decode color (turquoise) or width (black) from neural firing rate activity for each noise and regularization regime. Dotted gray lines and shading show mean ± 2 s.d. of chance level decoding based on shuffling trial labels 100 times. (**c**) Left: performance of optimized networks determined as the mean performance over all trained networks during the reward period (a, bottom; yellow shading) for all noise ($\sigma$, horizontal axis) and regularization levels ($\lambda$, vertical axis) used during training. Pale blue, pink, and purple dots indicate parameter values that correspond to the dynamical regimes shown in panel b and *Figure 1c* with the same background coloring. For parameter values above the white line, networks achieved a mean performance of less than 0.95. Middle: early color decoding determined as mean color decoding over all trained networks at the end of the color period (b, bottom left, turquoise arrow) using the same plotting scheme as the left panel. Right: width decoding determined as mean width decoding over all trained networks at the end of the shape period (b, bottom left, black arrow) using the same plotting scheme as the left panel. (**d**) Width decoding plotted against early color decoding for all noise and regularization levels and colored according to performance. Pale blue, pink, and purple highlights indicate the parameter values shown with the same colors in panel c.

The online version of this article includes the following figure supplement(s) for figure 2:

**Figure supplement 1.** Temporal decoding for networks with different noise and regularization levels.

## Stimulus representations in task-optimized recurrent neural networks

To understand the factors determining which representational strategy neural circuits employ to solve this task, we optimized recurrent neural networks to perform the task (*Figure 2a*; see also Neural network models). The neural activities in these stochastic recurrent networks evolved according to

$$\tau \frac{d\boldsymbol{x}(t)}{dt} = -\boldsymbol{x}(t) + \mathbf{W}\mathbf{r}(t) + \mathbf{h}(t) + \mathbf{b} + \sigma \boldsymbol{\eta}(t) \tag{1}$$

$$\text{with } \mathbf{r}(t) = \left[\boldsymbol{x}(t)\right]_+ \tag{2}$$

where $\boldsymbol{x}(t) = (x_1(t),\ldots,x_N(t))^\top$ corresponds to the vector of 'sub-threshold' activities of the $N = 50$ neurons in the network, $\mathbf{r}(t)$ is their momentary firing rates which is a rectified linear function (ReLU) of the sub-threshold activity, $\tau$=50 ms is the effective time constant, and $\mathbf{h}(t)$ denotes the inputs to the network encoding the three stimulus features as they become available over the course of the trial (*Figure 2a*, bottom). The optimized parameters of the network were $\mathbf{W}$, the recurrent weight matrix describing connection strengths between neurons in the network (*Figure 2a*, top; middle), $\mathbf{W}_{\mathrm{in}}$, the feedforward weight matrix describing connections from the stimulus inputs to the network (*Figure 2a*, top; left; see also Neural network models), and $\mathbf{b}$, a stimulus-independent bias. Importantly, $\sigma$ is the standard deviation of the neural noise process (*Figure 2a*, top; pale gray arrows; with $\boldsymbol{\eta}(t)$ being a sample from a Gaussian white noise process with mean 0 and variance 1), and as such represents a fundamental constraint on the operation of the network. The output of the network was given by

$$\mathbf{z}(t) = \mathrm{Softmax}\left(\mathbf{W}_{\mathrm{out}}\mathbf{r}(t) + \mathbf{b}_{\mathrm{out}}\right) \tag{3}$$

with optimized parameters $\mathbf{W}_{\mathrm{out}}$, the readout weights (*Figure 2a*, top; right), and $\mathbf{b}_{\mathrm{out}}$, a readout bias.

We optimized networks for a canonical cost function (*Orhan and Ma, 2019*; *Driscoll et al., 2022*; *Song et al., 2016*; *Stroud et al., 2021*; *Masse et al., 2019*; *Figure 2a*, bottom; Network optimization):

$$\mathcal{L} = \int_{t\in T_r} \mathcal{H}\left(\mathbf{c}, \mathbf{z}(t)\right) + \frac{\lambda}{2}\int_{t\in T} \|\mathbf{r}(t)\|_2^2 \tag{4}$$

The first term in *Equation 4* is a task performance term measuring the cross-entropy loss $\mathcal{H}(\mathbf{c},\mathbf{z}(t))$ between the one-hot encoded target choice, $\boldsymbol{c}$, and the network's output probabilities, $\mathbf{z}(t)$, during the reward period, $T_r$ (*Figure 2a*, bottom; yellow shading). The second term in *Equation 4* is a firing rate regularization term. This regularization term can be interpreted as a form of energy or metabolic cost (*Whittington et al., 2022*; *Kao et al., 2021*; *Cueva et al., 2020*; *Masse et al., 2019*; *Schimel et al., 2023*) measured across the whole trial, $T$, because it penalizes large overall firing rates. Therefore, optimizing this cost function encourages networks to not only solve the task, but to do so using low overall firing rates. How important it is for the network to keep firing rates low is controlled by the 'regularization' parameter $\lambda$. Critically, we used different noise levels $\sigma$ and regularization strengths $\lambda$ (*Figure 2a*, highlighted in red) to examine how these two constraints affected the dynamical strategies employed by the optimized networks to solve the task and compared them to our four hypothesized representational strategies (*Figure 1c*). We focused on these two parameters because they are both critical factors for real neural circuits and a priori can be expected to have important effects on the resulting circuit dynamics. For example, metabolic costs will constrain the overall level of firing rates that can be used to solve the task while noise levels directly affect how reliably stimuli can be encoded in the network dynamics. For the remainder of this section, we analyze representational strategies utilized by networks after training. In subsequent sections, we analyze learning-related changes in both our optimized networks and neural recordings.

We found that networks trained in a low noise–low firing rate regularization setting (which we denote by $\sigma_l, \lambda_l$) employed a maximal representational strategy (*Figure 2b*, left column; pale blue shading). Trajectories in neural firing rate space diverged for the two different colors as soon as the color stimulus was presented (*Figure 2b*, top left; blue and green trajectories from open gray circle to gray squares), which resulted in high color decoding during the color period (*Figure 2b*, bottom left; turquoise line; Linear decoding). During the shape period, the trajectories corresponding to each color diverged again (*Figure 2b*, top left; trajectories from gray squares to filled gray circles), such that all stimuli were highly decodable, including width (*Figure 2b*, bottom left; black line, *Figure 2—figure supplement 1a*). After removal of the stimuli during the reward period, trajectories converged to one of two parts of state space according to the XOR task rule – which is required for high performance (*Figure 2b*, top left; trajectories from filled gray circles to colored squares and diamonds). Because early color and width were highly decodable in these networks trained with a low noise and low firing rate regularization, the dynamical strategy they employed corresponds to the maximal representational regime (*Figure 1c*, blue shading).

We next considered the setting of networks that solve this task while being exposed to a high level of neural noise (which we denote by $\sigma_h, \lambda_l$; *Figure 2b*, middle column; purple shading). In this setting, we also observed neural trajectories that diverged during the color period (*Figure 2b*, top middle;

gray squares are separated), which yielded high color decoding during the color period (*Figure 2b*, bottom middle; turquoise line). However, in contrast to networks with a low level of neural noise (*Figure 2b*, left), width was poorly decodable during the shape period (*Figure 2b*, bottom middle; black line). Therefore, for networks challenged with higher levels of neural noise, the irrelevant stimulus dimension of width is represented more weakly (*Figure 1c*, purple shading). A similar representational strategy was also observed in networks that were exposed to a low level of noise but a high level of firing rate regularization (*Figure 2—figure supplement 1c*, black lines).

Finally, we considered the setting of networks that solve this task while being exposed to medium levels of both neural noise and firing rate regularization (which we denote by $\sigma_m, \lambda_m$; *Figure 2b*, right column; pink shading). In this setting, neural trajectories diverged only weakly during the color period (*Figure 2b*, top middle; gray squares on-top of one another), which yielded relatively poor (non-ceiling) color decoding during the color period (*Figure 2b*, bottom right; turquoise line). Nevertheless, color decoding was still above chance despite the neural trajectories strongly overlapping in the two-dimensional state space plot in *Figure 2b*, top right, because these trajectories became separable in the full-dimensional state space of these networks. Additionally, width decoding was also poor during the shape period (*Figure 2b*, bottom right; black line). Therefore, networks that were challenged with increased levels of both neural noise and firing rate regularization employed dynamics in line with a minimal representational strategy by only weakly representing irrelevant stimuli (*Figure 1c*, pink shading).

To gain a more comprehensive picture of the full range of dynamical solutions that networks can exhibit, we performed a grid search over multiple different levels of noise and firing rate regularization. Nearly all parameter values allowed networks to achieve high performance (*Figure 2c*, left), except when both the noise and regularization levels were high (*Figure 2c*, left; parameter values above white line). Importantly, all of the dynamical regimes that we showed in *Figure 2b* achieved similarly high performances (*Figure 2c*, left; pale blue, purple, and pink dots).

When looking at early color decoding (defined as color decoding at the end of the color period; *Figure 2b*, bottom left, turquoise arrow) and width decoding (defined as width decoding at the end of the shape period; *Figure 2b*, bottom left, black arrow), we saw a consistent pattern of results. Early color decoding was high only when either the lowest noise level was used (*Figure 2c*, middle; $\sigma = 0.01$ column) or when the lowest regularization level was used (*Figure 2c*, middle; $\lambda = 0.0005$ row). In contrast, width decoding was high only when both the level of noise and regularization were small (*Figure 2c*, right; bottom left corner). Otherwise, width decoding became progressively worse as either the noise or regularization level was increased and achieved values that were typically lower than the corresponding early color decoding (*Figure 2c*, compare right with middle). This pattern becomes clearer when we plot width decoding against early color decoding (*Figure 2d*). No set of parameters yielded higher width decodability compared to early color decodability (*Figure 2d*, no data point above the identity line). This means that we never observed the fourth dynamical regime we hypothesized a priori, in which width decoding would be high and early color decoding would be low (*Figure 1c*, top left). Therefore, information with static irrelevance (width) was more strongly suppressed compared to information whose irrelevance was dynamic (color). We also note that we never observed pure chance levels of decoding of color or width during stimulus presentation. This is likely because it is challenging for recurrent neural networks to strongly suppress their inputs and it may also be the case that other hyperparameter regimes more naturally lead to stronger suppression of inputs (we discuss this second possibility later; e.g. Figure 5).

## Comparing learning-related changes in stimulus representations in neural networks and primate lateral PFC

To understand the dynamical regime employed by PFC, we analyzed a dataset (*Wójcik, 2023*) of multi-channel recordings from lateral prefrontal cortex (lPFC) in two monkeys exposed to the task shown in *Figure 1a*. These recordings yielded 376 neurons in total across all recording sessions and both animals (Experimental materials and methods). Importantly, for understanding the direction in which neural geometries changed over learning, recordings commenced in the first session in which the animals were exposed to the task – i.e., the recordings spanned the entirety of the learning process. For our analyses, we distinguished between the first half of recording sessions (*Figure 3a*, gray; 'early learning') and the second half of recording sessions (*Figure 3a*, black; 'late learning'). A

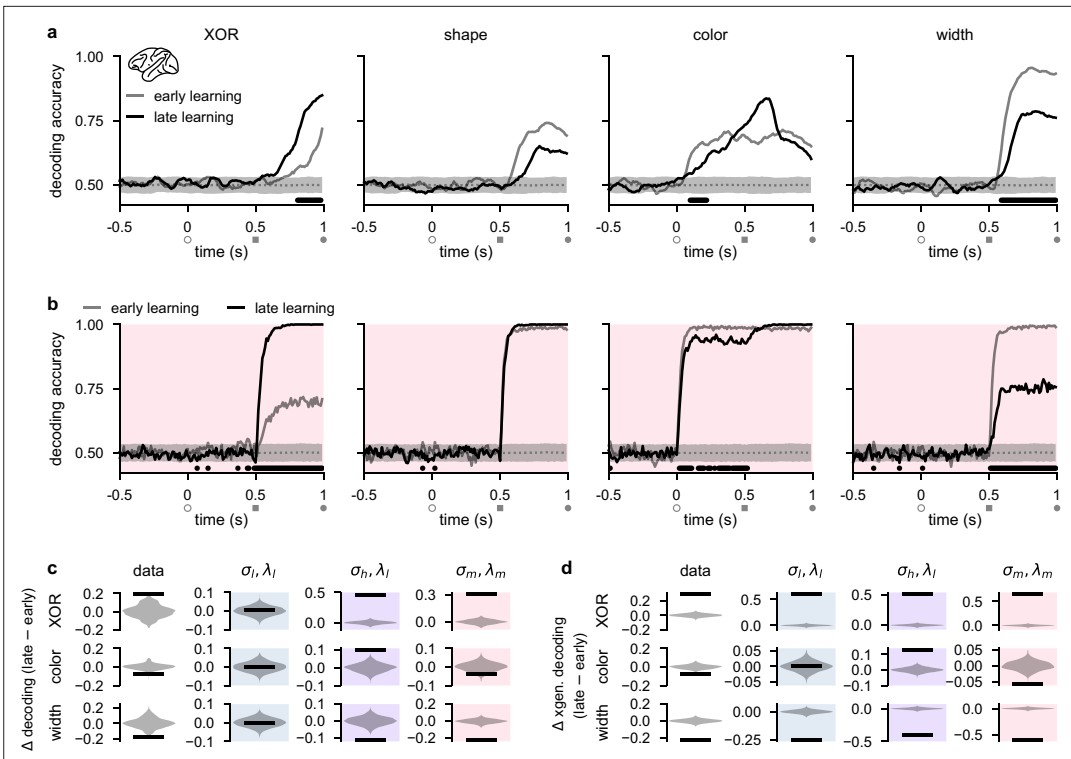

**Figure 3.** Stimulus representations in primate lateral prefrontal cortex (lPFC) correspond to a minimal representational strategy. (**a**) Performance of a linear decoder trained at each time point to predict either XOR (far left), shape (middle left), color (middle right), or width (far right) from neural population activity in lPFC (Experimental materials and methods and Linear decoding). We show decoding results separately from the first half of sessions (gray, 'early learning') and the second half of sessions (black, 'late learning'). Dotted gray lines and shading show mean ± 2 s.d. of chance level decoding based on shuffling trial labels 100 times. Horizontal black bars show significant differences between early and late decoding using a two-sided cluster-based permutation test and a significance threshold of 0.05 (Statistics). Open gray circles, filled gray squares, and filled gray circles on the horizontal indicate color onset, shape onset, and offset of both stimuli, respectively. (**b**) Same as panel a but for decoders trained on neural activity from optimized recurrent neural networks in the $\sigma_m, \lambda_m$ regime (*Figure 2b–d*, pink). (**c**) Black horizontal lines show the mean change between early and late decoding during time periods when there were significant differences between early and late decoding in the data (horizontal black bars in panel a) for XOR (top row), color (middle row), and width (bottom row). (No significant differences in shape decoding were observed in the data; cf. panel a.) Violin plots show chance differences between early and late decoding based on shuffling trial labels 100 times. We show results for the data (far left column), $\sigma_l, \lambda_l$ networks (middle left column, pale blue shading), $\sigma_h, \lambda_l$ networks (middle right column, purple shading), and $\sigma_m, \lambda_m$ networks (far right column, pink shading). (**d**) Same as panel c but we show results using cross-generalized decoding (*Bernardi et al., 2020*) during the same time periods as those used in panel c (Linear decoding).

The online version of this article includes the following figure supplement(s) for figure 3:

**Figure supplement 1.** Cross-generalized temporal decoding for lateral prefrontal cortex (lPFC) recordings and networks with different noise and regularization levels.

**Figure supplement 2.** Temporal decoding for networks with input weights initialized to 0 prior to optimization.

**Figure supplement 3.** Temporal decoding for networks with input weights initialized to large random values prior to optimization.

previous analysis of this dataset showed that, over the course of learning, the XOR representation of the task comes to dominate the dynamics during the late shape period of the task (*Wójcik, 2023*). Here, however, we focus on relevant and irrelevant task variable coding during the stimulus periods and compare the recordings to the dynamics of task-optimized recurrent neural networks. Also, in line with the previous study (*Wójcik, 2023*), we do not examine the reward period of the task because the one-to-one relationship between XOR and reward in the data (which is not present in the models) will

likely lead to trivial differences in XOR representations in the reward period between the data and models.

Similar to our analyses of our recurrent network models (*Figure 2b*, bottom), we performed population decoding of the key task variables in the experimental data (*Figure 3a*, see also Linear decoding). We found that the decodability of the XOR relationship between task-relevant stimuli that determined task performance (*Figure 1b*) significantly increased during the shape period over the course of learning, consistent with the animals becoming more familiar with the task structure (*Figure 3a*, far left; compare gray and black lines). We also found that shape decodability during the shape period decreased slightly, but not significantly, over learning (*Figure 3a*, middle left; compare gray and black lines from gray square to gray circle), while color decodability during the shape period increased slightly (*Figure 3a*, middle right; compare gray and black lines from gray square to gray circle). Importantly, however, color decodability significantly *decreased* during the color period (when it is 'irrelevant'; *Figure 3a*, middle right; compare gray and black lines), and width decodability significantly decreased during the shape period (*Figure 3a*, far right; compare gray and black lines). Neural activities in lPFC thus appear to change in line with the minimal representational strategy over the course of learning, consistent with recurrent networks trained in the $\sigma_m, \lambda_m$ regime (*Figure 1c*, pink and *Figure 2b*, bottom, pink).

We then directly compared these learning-related changes in stimulus decodability from lPFC with those that we observed in our task-optimized recurrent neural networks (*Figure 2*). We found that the temporal decoding dynamics of networks trained with medium noise and firing rate regularization ($\sigma_m, \lambda_m$; *Figure 2b–d*, pink shading and pink dots) exhibited decoding dynamics most similar to those that we observed in lPFC. Specifically, XOR decodability significantly increased after shape onset, consistent with the networks learning the task (*Figure 3b*, far left; compare gray and black lines). We also found that shape and color decodability did not significantly change during the shape period (*Figure 3b*, middle left and middle right; compare gray and black lines from gray square to gray circle). Importantly, however, color decodability significantly *decreased* during the color period (when it is 'irrelevant'; *Figure 3b*, middle right; compare gray and black lines from open gray circle to gray square), and width decodability significantly decreased during the shape period (*Figure 3a*, far right; compare gray and black lines). Other noise and regularization settings yielded temporal decoding that displayed a poorer resemblance to the data (*Figure 2—figure supplement 1*). For example, $\sigma_l, \lambda_l$, and $\sigma_l, \lambda_h$ networks exhibited almost no changes in decodability during the color and shape periods (*Figure 2—figure supplement 1a and c*) and $\sigma_h, \lambda_l$ networks exhibited increased XOR, shape, and color decodability at all times after stimulus onset while width decodability decreased during the shape period (*Figure 2—figure supplement 1b*). We also found that if regularization is driven to very high levels, color and shape decoding becomes weak during the shape period while XOR decoding remains high (*Figure 2—figure supplement 1d*). Therefore, such networks effectively perform a pure XOR computation during the shape period.

We also note that the $\sigma_m, \lambda_m$ model does not perfectly match the decoding dynamics seen in the data. For example, although not significant, we observed a decrease in shape decoding and an increase in color decoding in the data during the shape period whereas the model only displayed a slight (non-significant) increase in decodability of both shape and color during the same time period. These differences may be due to fundamentally different ways that brains encode sensory information upstream of PFC, compared to the more simplistic abstract sensory inputs used in models (see Discussion).

To systematically compare learning-related changes in the data and models, we analyzed time periods when there were significant changes in decoding in the data over the course of learning (*Figure 3a*, horizontal black bars). This yielded a substantial increase in XOR decoding during the shape period, and substantial decreases in color and width decoding during the color and shape periods, respectively, in the data (*Figure 3c*, far left column). During the same time periods in the models, networks in the $\sigma_l, \lambda_l$ regime exhibited no changes in decoding (*Figure 3c*, middle left column; blue shading). In contrast, networks in the $\sigma_h, \lambda_l$ regime exhibited substantial increases in XOR and color decoding, and a substantial decrease in width decoding (*Figure 3c*, middle right column; purple shading). Finally, in line with the data, networks in the $\sigma_m, \lambda_m$ regime exhibited a substantial increase in XOR decoding, and substantial decreases in both color and width decoding (*Figure 3c*, middle right column; pink shading).

In addition to studying changes in traditional decoding, we also studied learning-related changes in 'cross-generalized decoding' which provides a measure of how factorized the representational geometry is across stimulus conditions (*Bernardi et al., 2020*; *Figure 3—figure supplement 1*). (For example, for evaluating cross-generalized decoding for color, a color decoder trained on square trials would be tested on diamond trials, and vice versa *Bernardi et al., 2020*; see also Linear decoding.) Using this measure, we found that changes in decoding were generally more extreme over learning and that models and data bore a stronger resemblance to one another compared with traditional decoding. Specifically, all models and the data showed a strong increase in XOR decoding (*Figure 3d*, top row, 'XOR') and a strong decrease in width decoding (*Figure 3d*, bottom row, 'width'). However, only the data and $\sigma_m, \lambda_m$ networks showed a decrease in color decoding (*Figure 3d*, compare middle far left and middle far right), whereas $\sigma_l, \lambda_l$ networks showed no change in color decoding (*Figure 3d*, middle left; blue shading) and $\sigma_h, \lambda_l$ networks showed an increase in color decoding (*Figure 3d*, middle right; purple shading).

Beside studying the effects of input noise and firing rate regularization, we also examined the effects of different strengths of the initial stimulus input connections prior to training (*Flesch et al., 2022*). In line with previous studies (*Yang et al., 2019*; *Orhan and Ma, 2019*; *Stroud et al., 2021*; *Masse et al., 2019*; *Dubreuil et al., 2022*), for all of our previous results, the initial input weights were set to small random values prior to training (Network optimization). We found that changing these weights had similar effects to changing neural noise (with the opposite sign). Specifically, when input weights were set to 0 before training, initial decoding was at chance levels and only increased with learning for XOR, shape, and color (whereas width decoding hardly changed with learning and remained close to chance levels; *Figure 3—figure supplement 2*) – analogously to the $\sigma_h, \lambda_l$ regime (*Figure 2—figure supplement 1b*). In contrast, when input weights were set to large random values prior to training, initial decoding was at ceiling levels and did not significantly change over learning during the color and shape periods for any of the task variables (*Figure 3—figure supplement 3*) – similar to what we found in the $\sigma_l, \lambda_l$ regime (*Figure 2—figure supplement 1a*). Thus, neither extremely small nor extremely large initial input weights were consistent with the data that exhibited both increases and decreases in decodability of task variables over learning (*Figure 3a*).

## Theoretical predictions for strength of relevant and irrelevant stimulus coding

To gain a theoretical understanding of how irrelevant stimuli should be processed in a neural circuit, we performed a mathematical analysis of a minimal linear model that only included a single relevant stimulus and a single statically irrelevant stimulus (see also Appendix 1, Mathematical analysis of relevant and irrelevant stimulus coding in a linear network). Although this analysis applies to a simpler task compared with that faced by our neural networks and animals, crucially it still allows us to understand how relevant and irrelevant coding depend on noise and metabolic constraints. Our mathematical analysis suggested that the effects of noise and firing rate regularization on the performance and metabolic cost of a network can be understood via three key aspects of its representation: the strength of neural responses to the relevant stimulus ('relevant coding'), the strength of responses to the irrelevant stimulus ('irrelevant coding'), and the overlap between the population responses to relevant and irrelevant stimuli ('overlap'; *Figure 4a*). In particular, maximizing task performance (i.e. the decodability of the relevant stimulus) required relevant coding to be strong, irrelevant coding to be weak, and the overlap between the two to be small (*Figure 4b*, top; see also Appendix 1, The performance of the optimal linear decoder). This ensures that the irrelevant stimulus interferes minimally with the relevant stimulus. In contrast, to reduce a metabolic cost (such as we considered previously in our optimized recurrent networks, see *Equation 4* and *Figure 2*), both relevant and irrelevant coding should be weak (*Figure 4b*, bottom; see also Appendix 1, Metabolic cost).

In combination, when decoding performance and metabolic cost are *jointly* optimized, as in our task-optimized recurrent networks (*Figure 2*), our theoretical analyses suggested that performance should decrease with both the noise level $\sigma$ and the strength of firing rate regularization $\lambda$ in an approximately interchangeable way, and metabolic cost should increase with $\sigma$ but decrease with $\lambda$ (Appendix 1, Qualitative predictions about optimal parameters). We also found that the strength of relevant coding should decrease with $\lambda$, but its dependence on $\sigma$ was more nuanced. For small $\sigma$, the performance term effectively dominates the metabolic cost (*Figure 4b*, top) and the strength

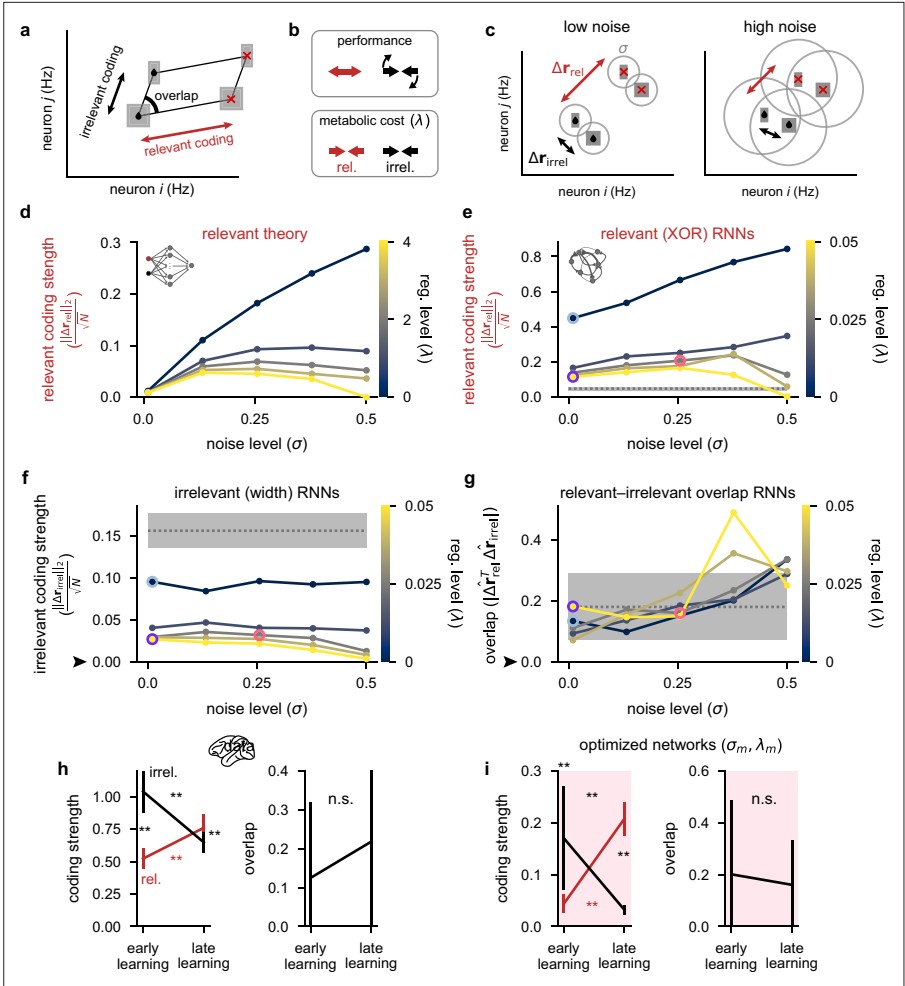

**Figure 4.** Theoretical predictions for strength of relevant and irrelevant stimulus coding and comparison to lateral prefrontal cortex (lPFC) recordings. (**a**) Schematic of activity in neural state space for two neurons for a task involving two relevant (black drops vs. red crosses) and two irrelevant stimuli (thick vs. thin squares). Strengths of relevant and irrelevant stimulus coding are shown with red and black arrows, respectively, and the overlap between relevant and irrelevant coding is also shown ('overlap'). (**b**) Schematic of our theoretical predictions for the optimal setting of relevant (red arrows) and irrelevant (black arrows) coding strengths when either maximizing performance (top) or minimizing a metabolic cost with strength $\lambda$ (bottom; see **Equation 4**). (**c**) Schematic of our theoretical predictions for the strength of relevant ($\Delta\mathbf{r}_{\mathrm{rel}}$; black drops vs. red crosses; red arrows) and irrelevant ($\Delta\mathbf{r}_{\mathrm{irrel}}$; thick vs. thin squares; black arrows) coding when jointly optimizing for both performance and a metabolic cost (cf. panel b). In the low noise regime (left), relevant conditions are highly distinguishable and irrelevant conditions are poorly distinguishable as well as strongly orthogonal to the relevant conditions. In the high noise regime (right), all conditions are poorly distinguishable. (**d**) Our theoretical predictions (**Equation S34**) for the strength of relevant coding ($\frac{\|\Delta\mathbf{r}_{\mathrm{rel}}\|_2}{\sqrt{N}}$, see panel c) as a function of the noise level $\sigma$ (horizontal axis) and firing rate regularization strength $\lambda$ (colorbar). (**e**) Same as panel d but for our optimized recurrent neural networks (**Figure 2**) where we show the strength of relevant (XOR) coding (Measuring stimulus coding strength). Pale blue, purple, and pink highlights correspond to the noise and regularization strengths shown in **Figure 2c and d**. Gray dotted line and shading shows mean ±2 s.d. (over 250 networks; 10 networks for each of the 25 different noise and regularization levels) of $\frac{\|\Delta\mathbf{r}_{\mathrm{rel}}\|_2}{\sqrt{N}}$ prior to training. (**f**) Same as panel e but for the strength of irrelevant (width) coding ($\frac{\|\Delta\mathbf{r}_{\mathrm{irrel}}\|_2}{\sqrt{N}}$). The black arrow indicates the theoretical prediction of 0 irrelevant coding. (**g**) The absolute value of the normalized dot product (overlap) between relevant and irrelevant representations ($\frac{|\Delta\mathbf{r}_{\mathrm{rel}}^{\top}\Delta\mathbf{r}_{\mathrm{irrel}}|}{\|\Delta\mathbf{r}_{\mathrm{rel}}\|_2\|\Delta\mathbf{r}_{\mathrm{irrel}}\|_2}$, i.e. 0 implies perfect orthogonality and 1 implies perfect overlap) for our optimized recurrent neural networks. The black arrow indicates the theoretical prediction of 0 overlap. (**h**) Left: coding strength (length of arrows in panel a; Measuring stimulus coding strength) for relevant (XOR; red) and irrelevant (width; black) stimuli during early and late learning for our lPFC recordings. Right: the absolute value of the overlap between relevant and irrelevant representations for our lPFC recordings (0 implies perfect orthogonality and 1 implies perfect overlap). Error bars show the mean ± 2 s.d.

*Figure 4 continued on next page*

*Figure 4 continued*

over 10 non-overlapping splits of the data. (**i**) Same as panel h but for the optimized recurrent neural networks in the $\sigma_m, \lambda_m$ regime (see pink dots in *Figure 2*). Error bars show the mean ± 2 s.d. over 10 different networks. p-Values resulted from a two-sided Mann–Whitney U test (\*, p<0.05; \*\*, p<0.01; n.s., not significant; see Statistics).

of relevant coding should increase with $\sigma$. However, if $\sigma$ is too high, the strength of relevant coding starts decreasing otherwise a disproportionately high metabolic cost must be incurred to achieve high performance (*Figure 4c,d*). Our mathematical analysis also suggested that irrelevant coding and relevant–irrelevant overlap could in principle depend on the noise and metabolic cost strength – particularly if performing noisy optimization where the curvature of the cost function can be relatively shallow around the minimum (Appendix 1, Qualitative predictions about optimal parameters). Therefore, practically, we also expect that irrelevant coding should be mostly dependent (inversely) on $\lambda$, but relevant–irrelevant overlap should mostly depend on $\sigma$ (Appendix 1, Curvature of the loss function around the optimum). These theoretical predictions were confirmed by our recurrent neural network simulations (*Figure 4e–g*).

We next measured, in both recorded and simulated population responses, the three aspects of population responses that our theory identified as being key in determining the performance and metabolic cost of a network (*Figure 4a*; Measuring stimulus coding strength). We found a close correspondence in the learning-related changes of these measures between our lPFC recordings (*Figure 4h*) and optimized recurrent networks (*Figure 4i*). In particular, we found that the strength of relevant (XOR) coding increased significantly over the course of learning (*Figure 4h and i*, left; red). The strength of irrelevant (width) coding decreased significantly over the course of learning (*Figure 4h and i*, left; black), such that it became significantly smaller than the strength of relevant coding (*Figure 4h and i*, left; compare red and black at 'late learning'). Finally, relevant and irrelevant directions were always strongly orthogonal in neural state space, and the level of orthogonality did not significantly change with learning (*Figure 4h and i*, right). Although we observed no learning-related changes in overlap, it may be that for stimuli that are more similar than the relevant and irrelevant features we studied here (XOR and width), the overlap between these features may decrease over learning rather than simply remaining small.

## Activity-silent, sub-threshold dynamics lead to the suppression of dynamically irrelevant stimuli

While the representation of statically irrelevant stimuli can be suppressed by simply weakening the input connections conveying information about it to the network, the suppression of dynamically irrelevant stimuli requires a mechanism that alters the dynamics of the network (since this information ultimately needs to be used by the network to achieve high performance). To gain some intuition about this mechanism, we first analyzed two-neuron networks trained on the task. To demonstrate the basic mechanism of suppression of dynamically irrelevant stimuli (i.e., weak early color coding), we compared the $\sigma_l, \lambda_l$ (*Figure 5a*) and $\sigma_m, \lambda_m$ (*Figure 5b*) regimes for these networks, as these corresponded to the minimal and maximal amount of suppression of dynamically irrelevant stimuli (*Figure 2c*).

We examined trajectories of sub-threshold neural activity $x(t)$ (*Equation 1*) in the full two-neuron state space (*Figure 5a and b*, blue and green curves). We distinguished between the negative quadrant of state space, which corresponds to the rectified part of the firing rate nonlinearity (*Figure 5a and b*, bottom left gray quadrant; *Equation 2*), and the rest of state space. Importantly, if sub-threshold activity lies within the negative quadrant at some time point, both neurons in the network have zero firing rate and consequently a decoder cannot decode any information from the firing rate activity and the network exhibits no metabolic cost (*Equation 4*). Therefore, we reasoned that color inputs may drive sub-threshold activity so that it lies purely in the negative quadrant of state space so that no metabolic cost is incurred during the color period (akin to nonlinear gating *Flesch et al., 2022*; *Miller and Cohen, 2001*). Later in the trial, when these color inputs are combined with the shape inputs, activity may then leave the negative quadrant of state space so that the network can perform the task.

We found that for the $\sigma_l, \lambda_l$ regime, there was typically at least one set of stimulus conditions for which sub-threshold neural activity evolved outside the negative quadrant of state space during any task period (*Figure 5a*). Consequently, color was decodable to a relatively high level during the color

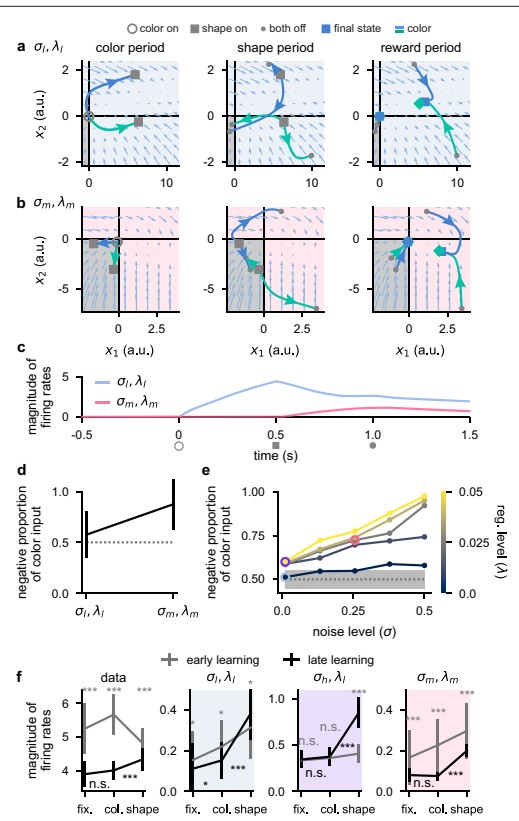

Figure 5 continued

and shading shows mean ± 2 s.d. (over 250 networks; 10 networks for each of the 25 different noise and regularization levels) of the proportion of negative color input prior to training (i.e. the proportion of negative color input expected when inputs are drawn randomly from a Gaussian distribution; Network optimization).

(**f**) Momentary magnitude of firing rates ($\frac{\|\mathbf{r}(t)\|_2}{\sqrt{N}}$) for our lateral prefrontal cortex (lPFC) recordings (far left) and 50-neuron networks in the $\sigma_l, \lambda_l$ regime (middle left, blue shading), $\sigma_h, \lambda_l$ regime (middle right, purple shading), and $\sigma_m, \lambda_m$ regime (far right, pink shading). Error bars show the mean ± 2 s.d. over either 10 non-overlapping data splits for the data or over 10 different networks for the models. p-Values resulted from a two-sided Mann–Whitney U test (*, p<0.05; **, p<0.01; ***, p<0.001; n.s., not significant; see Statistics).

The online version of this article includes the following figure supplement(s) for figure 5:

**Figure supplement 1.** Temporal decoding for two-neuron networks and supplemental analysis of suppression of dynamically irrelevant stimuli.

**Figure supplement 2.** Activity-silent, sub-threshold dynamics lead to the suppression of dynamically irrelevant stimuli in networks with a sigmoid nonlinearity.

**Figure 5.** Activity-silent, sub-threshold dynamics lead to the suppression of dynamically irrelevant stimuli. (**a**) Sub-threshold neural activity ($\mathbf{x}(t)$ in **Equation 1**) in the full state space of an example two-neuron network over the course of a trial (from color onset) trained in the $\sigma_l, \lambda_l$ regime. Pale blue arrows show flow field dynamics (direction and magnitude of movement in the state space as a function of the momentary state). Open gray circles indicate color onset, gray squares indicate shape onset, filled gray circles indicate offset of both stimuli, and colored squares and diamonds indicate the end of the trial at 1.5 s. We plot activity separately for the three periods of the task (color period, left; shape period, middle; reward period, right). We plot dynamics without noise for visual clarity. (**b**) Same as panel a but for a network trained in the $\sigma_m, \lambda_m$ regime. (**c**) Momentary magnitude of firing rates ($\frac{\|\mathbf{r}(t)\|_2}{\sqrt{N}}$; i.e. momentary metabolic cost, see **Equation 4**) for the two-neuron networks from panels a (blue line) and b (pink line). (**d**) Mean ± 2 s.d. (over 10 two-neuron networks) proportion of color inputs that have a negative sign for the two noise and regularization regimes shown in panels a–c. Gray dotted line shows chance level proportion of negative color input. (**e**) Mean (over 10 fifty-neuron networks) proportion of color inputs that have a negative sign for all noise (horizontal axis) and regularization (colorbar) strengths shown in **Figure 2c and d**. Pale blue, purple, and pink highlights correspond to the noise and regularization strengths shown in **Figure 2c and d**. Gray dotted line

*Figure 5 continued on next page*

period (**Figure 5—figure supplement 1a**, middle right) and this network produced a relatively high metabolic cost throughout the task (**Figure 5c**, blue line). In contrast, for networks in the $\sigma_m, \lambda_m$ regime, the two color inputs typically drove neural activity into the negative quadrant of state space during the color period (**Figure 5b**, left). Therefore, during the color period, the network produced zero firing rate activity (**Figure 5c**, pink line from open gray circle to gray square). Consequently, color was poorly decodable (**Figure 5—figure supplement 1b**, middle right; black line) and the network incurred no metabolic cost during the color period. Thus, color information was represented in a sub-threshold, 'activity-silent' (**Epsztein et al., 2011**) state during the color period. However, during the shape and reward periods later in the trial, the color inputs, now in combination with the shape inputs, affected the firing rate dynamics and the neural trajectories explored the full state space in a similar manner to the $\sigma_l, \lambda_l$ network (**Figure 5b**, middle and right panels). Indeed, we also found that color decodability increased substantially during the shape period in the $\sigma_m, \lambda_m$ network (**Figure 5—figure supplement 1b**, middle right; black line). This demonstrates how color inputs can cause no change in firing rates during the color period when they are irrelevant, and yet these same inputs can be

utilized later in the trial to enable high task performance (*Figure 5—figure supplement 1a and b*, far left). While we considered individual example networks in *Figure 5a–c*, color inputs consistently drove neural activity into the negative quadrant of state space across repeated training of $\sigma_m, \lambda_m$ networks but not for $\sigma_l, \lambda_l$ networks (*Figure 5d*).

Next, we performed the same analysis as *Figure 5d* on the large (50-neuron) networks that we studied previously (*Figures 2–4*). Similar to the two-neuron networks, we found that large networks in the $\sigma_l, \lambda_l$ regime did not exhibit more negative color inputs than would be expected by chance (*Figure 5e*, pale blue highlighted point) – consistent with the high early color decoding we found previously in these networks (*Figure 2c*, middle, pale blue highlighted point). However, when increasing either the noise or regularization level, the proportion of negative color inputs increased above chance such that for the highest noise and regularization level, nearly all color inputs were negative (*Figure 5e*). We also found a strong negative correlation between the proportion of negative color input and the level of early color decoding that we found previously in *Figure 2c* (*Figure 5—figure supplement 1c*, $\mathrm{r} = -0.9, \mathrm{p} < 10^{-9}$). This suggests that color inputs that drive neural activity into the rectified part of the firing rate nonlinearity, and thus generate purely sub-threshold activity-silent dynamics, is the mechanism that generates weak early color coding during the color period in these networks. We also examined whether these results generalized to networks that use a sigmoid (as opposed to a ReLU) nonlinearity. To do this, we trained networks with a *tanh* nonlinearity (shifted for a meaningful comparison with ReLU, so that the lower bound on firing rates was 0, rather than –1) and found qualitatively similar results to the ReLU networks. In particular, color inputs drove neural activity toward 0 firing rate during the color period in $\sigma_m, \lambda_m$ networks but not in $\sigma_l, \lambda_l$ networks (*Figure 5—figure supplement 2*, compare a and b), which resulted in a lower metabolic cost during the color period for $\sigma_m, \lambda_m$ networks compared to $\sigma_l, \lambda_l$ networks (*Figure 5—figure supplement 2c*). This was reflected in color inputs being more strongly negative in $\sigma_m, \lambda_m$ networks compared to $\sigma_l, \lambda_l$ networks which only showed chance levels of negative color inputs (*Figure 5—figure supplement 2d*).

We next sought to test whether this mechanism could also explain the decrease in color decodability over learning that we observed in the lPFC data. To do this, we measured the magnitude of firing rates in the fixation, color, and shape periods for both early and late learning (note that the magnitude of firing rates coincides with our definition of the metabolic cost; *Equation 4* and *Figure 5c*). To decrease metabolic cost over learning, we would expect two changes: firing rates should decrease with learning and firing rates should not significantly increase from the fixation to the color period after learning (*Figure 5c*, pink line). Indeed, we found that firing rates decreased significantly over the course of learning in all task periods (*Figure 5f*, far left; compare gray and black lines), and this decrease was most substantial during the fixation and color periods (*Figure 5f*, far left). We also found that after learning, firing rates during the color period were not significantly higher than during the fixation period (*Figure 5f*, far left; compare black error bars during fixation and color periods). During the shape period however, firing rates increased significantly compared to those during the fixation period (*Figure 5f*, far left; compare black error bars during fixation and shape periods). Therefore, the late learning dynamics of the data are in line with what we saw for the optimized two-neuron $\sigma_m, \lambda_m$ network (*Figure 5c*, pink line).

We then compared these results from our neural recordings with the results from our large networks trained in different noise and regularization regimes. We found that for $\sigma_l, \lambda_l$ networks, over the course of learning, firing rates decreased slightly during the fixation and color periods but actually increased during the shape period (*Figure 5f*, middle left pale blue shading; compare gray and black lines). Additionally, after learning, firing rates increased between fixation and color periods and between color and shape periods (*Figure 5f*, middle left pale blue shading; compare black error bars). For $\sigma_h, \lambda_l$ networks, over the course of learning, we found that firing rates did not significantly change during the fixation and color periods but increased significantly during the shape period (*Figure 5f*, middle right purple shading; compare gray and black lines). Furthermore, after learning, firing rates did not change significantly between fixation and color periods but did increase significantly between color and shape periods (*Figure 5f*, middle right purple shading; compare black error bars). Finally, for the $\sigma_m, \lambda_m$ networks, we found a pattern of results that was most consistent with the data. Firing rates decreased over the course of learning in all task periods (*Figure 5f*, far right pink shading; compare gray and black lines) and the decrease in firing rates was most substantial during the fixation and color

periods. After learning, firing rates did not change significantly from fixation to color periods but did increase significantly during the shape period (*Figure 5f*, far right; compare black error bars).

To investigate whether these findings depended on our random network initialization prior to training, we also compared late learning firing rates to firing rates that resulted from randomly shuffling the color, shape, and width inputs (which emulates alternative tasks where different combinations of color, shape, and width are relevant). For example, the relative strengths of the three inputs to the network prior to training on this task may affect how the firing rates change over learning. Under this control, we also found that $\sigma_m$, $\lambda_m$ networks bore a close resemblance to the data (*Figure 5—figure supplement 1d*, compare black lines to blue lines).

## Discussion

Comparing the neural representations of task-optimized networks with those observed in experimental data has been particularly fruitful in recent years (*Wang et al., 2018*; *Mante et al., 2013*; *Sussillo et al., 2015*; *Barak et al., 2013*; *Cueva et al., 2020*; *Echeveste et al., 2020*; *Stroud et al., 2021*; *Lindsay, 2021*). However, networks are typically optimized using only a very limited range of hyperparameter values (*Wang et al., 2018*; *Mante et al., 2013*; *Sussillo et al., 2015*; *Barak et al., 2013*; *Cueva et al., 2020*; *Driscoll et al., 2022*; *Song et al., 2016*; *Echeveste et al., 2020*; *Stroud et al., 2021*). Instead, here we showed that different settings of key, biologically relevant hyperparameters such as noise and metabolic costs, can yield a variety of qualitatively different dynamical regimes that bear varying degrees of similarity with experimental data. In general, we found that increasing levels of noise and firing rate regularization led to increasing amounts of irrelevant information being filtered out in the networks. Indeed, filtering out of task-irrelevant information is a well-known property of the PFC and has been observed in a variety of tasks (*Freedman et al., 2001*; *Duncan, 2001*; *Cueva et al., 2020*; *Reinert et al., 2021*; *Miller and Cohen, 2001*; *Rainer and Miller, 2002*; *Asaad et al., 2000*; *Everling et al., 2002*). We provide a mechanistic understanding of the specific conditions that lead to stronger filtering of task-irrelevant information. We predict that these results should also generalize to richer, more complex cognitive tasks that may, for example, require context switching (*Flesch et al., 2022*; *Reinert et al., 2021*; *Asaad et al., 2000*) or invoke working memory (*Freedman et al., 2001*; *Rainer et al., 1998*; *Asaad et al., 2000*). Indeed, filtering out of task-irrelevant information in the PFC has been observed in such tasks (*Freedman et al., 2001*; *Rainer et al., 1998*; *Flesch et al., 2022*; *Reinert et al., 2021*; *Asaad et al., 2000*; *Mack et al., 2020*).

Our results are also likely a more general finding of neural circuits that extend beyond the PFC. In line with this, it has previously been shown that strongly regularized neural network models trained to reproduce monkey muscle activities during reaching bore a stronger resemblance to neural recordings from primary motor cortex compared to unregularized models (*Sussillo et al., 2015*). In related work on motor control, recurrent networks controlled by an optimal feedback controller recapitulated key aspects of experimental recordings from primary motor cortex (such as orthogonality between preparatory and movement neural activities) when the control input was regularized (*Kao et al., 2021*). Additionally, regularization of neural firing rates, and its natural biological interpretation as a metabolic cost, has recently been shown to be a key ingredient for the formation of grid cell-like response profiles in artificial networks (*Whittington et al., 2022*; *Cueva and Wei, 2018*).

By showing that PFC representations changed in line with a minimal representational strategy, our results are in line with various studies showing low-dimensional representations under a variety of tasks in the PFC and other brain regions (*Rainer et al., 1998*; *Flesch et al., 2022*; *Cueva et al., 2020*; *Ganguli et al., 2008*; *Sohn et al., 2019*). This is in contrast to several previous observations of high-dimensional neural activity in PFC (*Rigotti et al., 2013*; *Bernardi et al., 2020*). Both high- and low-dimensional regimes confer distinct yet useful benefits: high-dimensional representations allow many behavioral readouts to be generated, thereby enabling highly flexible behavior (*Rigotti et al., 2013*; *Flesch et al., 2022*; *Barak et al., 2013*; *Enel et al., 2016*; *Maass et al., 2002*; *Fusi et al., 2016*), whereas low-dimensional representations are more robust to noise and allow for generalization across different stimuli (*Flesch et al., 2022*; *Barak et al., 2013*; *Fusi et al., 2016*). These two different representational strategies have previously been studied in models by setting the initial network weights to either small values (to generate low-dimensional 'rich' representations) or large values (*Flesch et al., 2022*) (to generate high-dimensional 'lazy' representations). However, in contrast to previous approaches, we studied the more biologically plausible effects of firing rate regularization

(i.e. a metabolic cost; see also the supplement of *Flesch et al., 2022*) on the network activities over the course of learning and compared them to learning-related changes in PFC neural activities. Firing rate regularization will cause neural activities to wax and wane as networks are exposed to new tasks depending upon the stimuli that are currently relevant. In line with this, it is conceivable that prolonged exposure to a task that has a limited number of stimulus conditions, some of which can even be generalized over (as was the case for our task), encourages more low-dimensional dynamics to form (*Wójcik, 2023*; *Flesch et al., 2022*; *Dubreuil et al., 2022*; *Fusi et al., 2016*; *Musslick, 2017*; *Mastrogiuseppe and Ostojic, 2018*). In contrast, tasks that use a rich variety of stimuli (that may even dynamically change over the task *Wang et al., 2018*; *Jensen et al., 2023*; *Heald et al., 2021*), and which do not involve generalization across stimulus conditions, may more naturally lead to high-dimensional representations (*Rigotti et al., 2013*; *Barak et al., 2013*; *Dubreuil et al., 2022*; *Fusi et al., 2016*; *Mastrogiuseppe and Ostojic, 2018*; *Bartolo et al., 2020*). It would therefore be an important future question to understand how our results also depend on the task being studied as some tasks may more naturally lead to the 'maximal' representational regime (*Barak et al., 2013*; *Dubreuil et al., 2022*; *Mastrogiuseppe and Ostojic, 2018*; *Figures 1–3 and 5*, blue shading).

A previous analysis of the same dataset that we studied here focused on the late parts of the trial (*Wójcik, 2023*). In particular, they found that the final result of the computation needed for the task, the XOR operation between color and shape, emerges and eventually comes to dominate lPFC representations over the course of learning in the late shape period (*Wójcik, 2023*). Our analysis goes beyond this by studying mechanisms of suppression of both static and dynamically irrelevant stimuli across all task periods and how different levels of neural noise and metabolic cost can lead to qualitatively different representations of irrelevant stimuli in task-optimized recurrent networks. Other previous studies focused on characterizing the representation of several task-relevant (*Rigotti et al., 2013*; *Bernardi et al., 2020*; *Stokes et al., 2013*) – and, in some cases, -irrelevant (*Flesch et al., 2022*) – variables over the course of individual trials. Characterizing how key aspects of neural representations change over the course of learning, as we did here, offers unique opportunities for studying the functional objectives shaping neural representations (*Richards et al., 2019*).

There were several aspects of the data that were not well captured by our models. For example, during the shape period, decodability of shape decreased while decodability of color increased (although not significantly) in our neural recordings (*Figure 3a*). These differences in changes in decoding may be due to fundamentally different ways that brains encode sensory information upstream of PFC, compared to our models. For example, shape and width are both geometric features of the stimulus, whose encoding is differentiated from that of color already at early stages of visual processing (*Kandel, 2000*). Such a hierarchical representation of inputs may automatically lead to the (un)learning about the relevance of width (which the $\sigma_m$, $\lambda_m$ model reproduced) generalizing to shape, but not to color. In contrast, inputs in our model used a non-hierarchical one-hot encoding (*Figure 2*), which did not allow for such generalization. Moreover, in the data, we may expect width to be a priori more strongly represented than color or shape because it is a much more potent sensory feature. In line with this, in our neural recordings, we found that width was very strongly represented in early learning compared to the other stimulus features (*Figure 3a*, far right) and width always yielded high cross-generalized decoding – even after learning (*Figure 3—figure supplement 1a*, far right). Nevertheless, studying *changes* in decoding over learning, rather than absolute decoding levels, allowed us to focus on features of learning that do not depend on the details of upstream sensory representations of stimuli. Future studies could incorporate aspects of sensory representations that we ignored here by using stimulus inputs with which the model more faithfully reproduces the experimentally observed initial decodability of stimulus features.

In line with previous studies (*Yang et al., 2019*; *Whittington et al., 2022*; *Sussillo et al., 2015*; *Orhan and Ma, 2019*; *Kao et al., 2021*; *Cueva et al., 2020*; *Driscoll et al., 2022*; *Song et al., 2016*; *Stroud et al., 2021*; *Masse et al., 2019*; *Schimel et al., 2023*), we operationalized metabolic cost in our models through $L_2$ firing rate regularization. This cost penalizes high overall firing rates. There are however alternative conceivable ways to operationalize a metabolic cost; e.g., $L_1$ firing rate regularization has been used previously when optimizing neural networks and promotes more sparse neural firing (*Yang et al., 2019*). Interestingly, although our $L_2$ is generally conceived to be weaker than $L_1$ regularization, we still found that it encouraged the network to use purely sub-threshold activity in our task. The regularization of synaptic weights may also be biologically relevant (*Yang et al., 2019*)

because synaptic transmission uses the most energy in the brain compared to other processes (*Faria-Pereira and Morais, 2022*; *Harris et al., 2012*). Additionally, even sub-threshold activity could be regularized as it also consumes energy (although orders of magnitude less than spiking *Zhu et al., 2019*). Therefore, future work will be needed to examine how different metabolic costs affect the dynamics of task-optimized networks.

We build on several previous studies that have also analyzed learning-related changes in PFC activity (*Wójcik, 2023*; *Reinert et al., 2021*; *Durstewitz et al., 2010*; *Bartolo et al., 2020*) – although these studies typically used reversal-learning paradigms in which animals are already highly task proficient and the effects of learning and task switching are inseparable. For example, in a rule-based categorization task in which the categorization rule changed after learning an initial rule, neurons in mouse PFC adjusted their selectivity depending on the rule such that currently irrelevant information was not represented (*Reinert et al., 2021*). Similarly, neurons in rat PFC transition rapidly from representing a familiar rule to representing a completely novel rule through trial-and-error learning (*Durstewitz et al., 2010*). Additionally, the dimensionality of PFC representations was found to increase as monkeys learned the value of novel stimuli (*Bartolo et al., 2020*). Importantly, however, PFC representations did not distinguish between novel stimuli when they were first chosen. It was only once the value of the stimuli were learned that their representations in PFC were distinguishable (*Bartolo et al., 2020*). These results are consistent with our results where we found poor XOR decoding during the early stages of learning which then increased over learning as the monkeys learned the rewarded conditions (*Figure 3a*, far left). However, we also observed high decoding of width during early learning which was not predictive of reward (*Figure 3a*, far right). One key distinction between our study and that of *Bartolo et al., 2020*, is that our recordings commenced from the first trial the monkeys were exposed to the task. In contrast, in *Bartolo et al., 2020*, the monkeys were already highly proficient at the task and so the neural representation of their task was already likely strongly task specific by the time recordings were taken.

In line with our approach here, several recent studies have also examined the effects of different hyperparameter settings on the solution that optimized networks exhibit. One study found that decreasing regularization on network weights led to more sequential dynamics in networks optimized to perform working memory tasks (*Orhan and Ma, 2019*). Another study found that the number of functional clusters that a network exhibits does not depend strongly on the strength of ($L_1$ rate or weight) regularization, but did depend upon whether the single neuron nonlinearity saturates at high firing rates (*Yang et al., 2019*). It has also been shown that networks optimized to perform path integration can exhibit a range of different properties, from grid cell-like receptive fields to distinctly non grid cell-like receptive fields, depending upon biologically relevant hyperparameters – including noise and regularization (*Whittington et al., 2022*; *Cueva and Wei, 2018*; *Schaeffer et al., 2022*). Indeed, in addition to noise and regularization, various other hyperparameters have also been shown to affect the representational strategy used by a circuit, such as the firing rate nonlinearity (*Yang et al., 2019*; *Whittington et al., 2022*; *Schaeffer et al., 2022*) and network weight initialization (*Flesch et al., 2022*; *Schaeffer et al., 2022*). It is therefore becoming increasingly clear that analyzing the interplay between learning and biological constraints will be key for understanding the computations that various brain regions perform.

## Methods
### Experimental materials and methods

Experimental methods have been described previously (*Wójcik, 2023*). The experiments were conducted in line with the Animals (Scientific Procedures) Act 1986 of the UK and licensed by a Home Office Project License obtained after review by Oxford University's Animal Care and Ethical Review committee. The procedures followed the standards set out in the European Community for the care and use of laboratory animals (EUVD, European Union directive 86/609/EEC). Briefly, two adult male rhesus macaques (monkey 1 and monkey 2) performed a passive object–association task (*Figure 1a and b*; see the main text 'A task involving relevant and irrelevant stimuli' for a description of the task). Neural recordings commenced from the first session the animals were exposed to the task. All trials with fixation errors were excluded. The dataset contained on average 237.9 (s.d. = 23.9) and 104.8 (s.d. = 2.3) trials for each of the eight conditions for monkeys 1 and 2, respectively. Data were

recorded from the ventral and dorsal lPFC over a total of 27 daily sessions across both monkeys which yielded 146 and 230 neurons for monkey 1 and monkey 2, respectively. To compute neural firing rates, we convolved binary spike trains with a Gaussian kernel with a standard deviation of 50 ms. In order to characterize changes in neural dynamics over learning, analyses were performed separately on the first half of sessions ('early learning'; 9 and 5 sessions from monkey 1 and monkey 2, respectively) and the second half of sessions ('late learning'; 8 and 5 sessions from monkey 1 and monkey 2, respectively; *Figures 3–5* and *Figure 3—figure supplement 1a*). This experiment was only performed once in these two animals.

## Neural network models

The dynamics of our simulated networks evolved according to *Equations 1 and 2* and are repeated here:

$$\tau \frac{d\boldsymbol{x}\left(t\right)}{dt} = -\boldsymbol{x}\left(t\right) + \mathbf{W}\mathbf{r}\left(t\right) + \mathbf{h}\left(t\right) + \mathbf{b} + \sigma\boldsymbol{\eta}\left(t\right) \tag{5}$$

$$\text{with } \mathbf{r}\left(t\right) = \left[\boldsymbol{x}\left(t\right)\right]_{+} \tag{6}$$

$$\text{and } \mathbf{h}\left(t\right) = \mathbf{W}_{\text{in}}^{c}\mathbf{h}^{c}\left(t\right) + \mathbf{W}_{\text{in}}^{s}\mathbf{h}^{s}\left(t\right) + \mathbf{W}_{\text{in}}^{w}\mathbf{h}^{w}\left(t\right) \tag{7}$$

where $\boldsymbol{x}\left(t\right) = \left(x_1\left(t\right),\ldots,x_N\left(t\right)\right)^{\top}$ corresponds to the vector of 'sub-threshold' activities of the $N$ neurons in the network, $\mathbf{r}\left(t\right)$ is their momentary firing rates, obtained as a rectified linear function (ReLU) of their sub-threshold activities (*Equation 6*; except for the networks of *Figure 5—figure supplement 2* in which we used a *tanh* nonlinearity to examine the generalizability of our results), $\tau$=50 ms is the effective time constant, $\mathbf{W}$ is the recurrent weight matrix, $\mathbf{h}\left(t\right)$ denotes the total stimulus input, $\boldsymbol{b}$ is a stimulus-independent bias, $\sigma$ is the standard deviation of the neural noise process, $\boldsymbol{\eta}\left(t\right)$ is a sample from a Gaussian white noise process with mean 0 and variance 1, $\mathbf{W}_{\text{in}}^{c}, \mathbf{W}_{\text{in}}^{s}$, and $\mathbf{W}_{\text{in}}^{w}$ are color, shape, and width input weights, respectively, and $\mathbf{h}^{c}\left(t\right), \mathbf{h}^{s}\left(t\right)$, and $\mathbf{h}^{w}\left(t\right)$ are one-hot encodings of color, shape, and width inputs, respectively.

All simulations started at $t_0 = -0.5$ s and lasted until $t_{\text{max}} = 1.5$ s, and consisted of a fixation (–0.5 to 0 s), color (0–0.5 s), shape (0.5–1 s), and reward (1–1.5 s) period (*Figures 1a and 2a*). The initial condition of neural activity was set to $\boldsymbol{x}\left(t_0\right) = \mathbf{0}$. In line with the task, elements of $\mathbf{h}^{c}\left(t\right)$ were set to 0 outside the color and shape periods, and elements of both $\mathbf{h}^{s}\left(t\right)$ and $\mathbf{h}^{w}\left(t\right)$ were set to 0 outside the shape period. All networks used $N$=50 neurons (except for *Figure 5a–d* and *Figure 5—figure supplement 1a and b* which used $N$=2 neurons). We solved the dynamics of *Equations 5–7* using a first-order Euler–Maruyama approximation with a discretization time step of 1 ms.

### Network optimization

Choice probabilities were computed through a linear readout of network activities:

$$\mathbf{z}\left(t\right) = \text{Softmax}\left(\mathbf{W}_{\text{out}}\mathbf{r}\left(t\right) + \mathbf{b}_{\text{out}}\right) \tag{8}$$

where $\mathbf{W}_{\text{out}}$ are the readout weights and $\mathbf{b}_{\text{out}}$ is a readout bias. To measure network performance, we used a canonical cost function (*Yang et al., 2019*; *Orhan and Ma, 2019*; *Driscoll et al., 2022*; *Song et al., 2016*; *Stroud et al., 2021*; *Masse et al., 2019*; *Equations 3 and 4*). We repeat the cost function from the main text here:

$$\mathcal{L} = \int_{t=1.0}^{t=1.5} \mathcal{H}\left(\mathbf{c}, \mathbf{z}\left(t\right)\right) + \frac{\lambda}{2}\int_{t=-0.5}^{t=1.5} \left\|\mathbf{r}\left(t\right)\right\|_2^2 \tag{9}$$

where the first term is a task performance term which consists of the cross-entropy loss $\mathcal{H}\left(\mathbf{c}, \mathbf{z}\left(t\right)\right) = -\sum_{k=1}^{2} c_k \, ln\, z_k\left(t\right)$ between the one-hot encoded target choice, $\mathbf{c}$ (based on the stimuli of the given trial, as defined by the task rules, *Figure 1b*), and the network's readout probabilities, $\mathbf{z}\left(t\right)$ (*Equation 8*). Note that we measure total classification performance (cross-entropy loss) during the reward period (integral in the first term runs from $t$=1.0 to $t$=1.5; *Figure 2a*, bottom; yellow shading), as appropriate for the task. The second term in *Equation 9* is a widely used (*Yang et al., 2019*; *Whittington et al., 2022*; *Sussillo et al., 2015*; *Orhan and Ma, 2019*; *Kao et al., 2021*; *Driscoll et al., 2022*; *Song et al., 2016*; *Stroud et al., 2021*; *Masse et al., 2019*) $L_2$ regularization term (with strength

**Table 1.** Parameters used in the simulations of our models.

| Symbol | Figures 2 and 3b-d Figure 4e-g,i, Figure 5e,f Figure 2—figure supplement 1a-c, Figure 3—figure supplements 2 and 3, Figure 3—figure supplement 1b,c, and Figure 5—figure supplement 1c,d | Figure 5a-d and Figure 5—figure supplement 1a,b | Figure 5—figure supplement 2 | Figure 2—figure supplement 1d | Units | Description |
|---|---|---|---|---|---|---|
| $N$ | 50 | 2 | 2 | 50 | - | number of neurons |
| $t_0$ | −0.5 | −0.5 | −0.5 | −0.5 | s | simulation start time |
| $t_{\max}$ | 1.5 | 1.5 | 1.5 | 1.5 | s | simulation end time |
| $\tau$ | 0.05 | 0.05 | 0.05 | 0.05 | s | effective time constant |
| $r\left(\boldsymbol{x}\left(t\right)\right)$ | ReLU | ReLU | $tanh\left(\boldsymbol{x}\left(t\right)\right)+1$ | ReLU | Hz | nonlinearity |
| $\sigma$ | variable* | [0.01, 0.255] | [0.01, 0.255] | 0.01 | - | noise standard deviation |
| $\lambda$ | variable† | [0.0005, 0.02525] | [0.0005, 0.02525] | 0.5 | s | regularization strength |
| $\mathbf{W}$ | optimized ‡ | optimized ‡ | optimized ‡ | optimized ‡ | s | weight matrix |
| $\mathbf{W}_{\mathrm{in}}^c$ | optimized ‡ | optimized ‡ | optimized ‡ | optimized ‡ | - | color input weights |
| $\mathbf{W}_{\mathrm{in}}^s$ | optimized ‡ | optimized ‡ | optimized ‡ | optimized ‡ | - | shape input weights |
| $\mathbf{W}_{\mathrm{in}}^w$ | optimized ‡ | optimized ‡ | optimized ‡ | optimized ‡ | - | width input weights |
| $\mathbf{b}$ | optimized ‡ | optimized ‡ | optimized ‡ | optimized ‡ | - | bias |
| $\mathbf{W}_{\mathrm{out}}$ | optimized ‡ | optimized ‡ | optimized ‡ | optimized ‡ | s | readout weights |
| $\mathbf{b}_{\mathrm{out}}$ | optimized ‡ | optimized ‡ | optimized ‡ | optimized ‡ | - | readout bias |

\* The noise standard deviation $\sigma$ took one of 5 evenly spaced values between 0.01 and 0.5 (see **Figure 2c**).

†The firing rate regularization strength $\lambda$ took one of 5 evenly spaced values between 0.0005 and 0.05 (see **Figure 2c**).

‡ See Neural network models section for details.

$\lambda$) applied to the neural firing rates throughout the trial (integral in the second term runs from $t=-0.5$ to $t$=1.5).

We initialized the free parameters of the network (the elements of $\mathbf{W}_{\mathrm{in}}^c, \mathbf{W}_{\mathrm{in}}^s, \mathbf{W}_{\mathrm{in}}^w, \mathbf{W}, \mathbf{b}, \mathbf{W}_{\mathrm{out}}$, and $\mathbf{b}_{\mathrm{out}}$) by sampling (independently) from a normal distribution of mean 0 and standard deviation $1/\sqrt{N}$. There were two exceptions to this: we also investigated the effects of initializing the elements of the input weights ($\mathbf{W}_{\mathrm{in}}^c, \mathbf{W}_{\mathrm{in}}^s, \mathbf{W}_{\mathrm{in}}^w$) to either 0 (**Figure 3—figure supplement 2**) or sampling their elements from a normal distribution of mean 0 and standard deviation $10/\sqrt{N}$ (**Figure 3—figure supplement 3**). After initialization, we optimized these parameters using gradient descent with Adam (**Kingma and Ba, 2014**), where gradients were obtained from backpropagation through time. We used a learning rate of 0.001 and trained networks for 1000 iterations using a batch size of 10. For each noise $\sigma$ and regularization level $\lambda$ (see **Table 1**), we optimized 10 networks with different random initializations of the network parameters.

## Analysis methods

Here, we describe methods that we used to analyze neural activities. No data were excluded from our analyses. Whenever applicable, the same processing and analysis steps were applied to both experimentally recorded and model simulated data. All neural firing rates were sub-sampled at a 10 ms resolution and, unless stated otherwise, we did not trial-average firing rates before performing analyses. Analyses were either performed at every time point in the trial (**Figures 2b and 3a, b**, **Figure 5a–c**, **Figure 2—figure supplement 1**, **Figure 3—figure supplements 1–3**, and **Figure 5—figure supplement 1a and b**), at the end of either the color (**Figure 2c and d**, and **Figure 5—figure supplement**

*1c*; 'early color decoding') or shape periods (*Figure 2c and d*, 'width decoding'), during time periods of significant changes in decoding over learning in the data (*Figure 3c and d*), during the final 100 ms of the shape period (*Figure 4e–i*), or during the final 100 ms of the fixation, color, and shape periods (*Figure 5f* and *Figure 5—figure supplement 1d*).

## Linear decoding

For decoding analyses (*Figures 2b–d and 3*, *Figure 2—figure supplement 1*, *Figure 3—figure supplements 1–3*, and *Figure 5—figure supplement 1a–c*), we fitted decoders using linear support vector machines to decode the main task variables: color, shape, width, and the XOR between color and shape. We measured decoding performance in a cross-validated way, using separate sets of trials to train and test the decoders, and we show results averaged over 10 random 1:1 train:test splits. For firing rates resulting from simulated neural networks, we used 10 trials for both the train and test splits. Chance level decoding was always 0.5 as all stimuli were binary.

For showing changes in decoding over learning (*Figure 3c and d*), we identified time periods of significant changes in decoding during the color and shape periods in the data (*Figure 3a*, horizontal black bars; see Statistics), and show the mean change in decoding during these time periods for both the data and models (*Figure 3c*, horizontal black lines). We used the same time periods when showing changes in cross-generalized decoding over learning (*Figure 3d*, see below).

For cross-generalized decoding (*Bernardi et al., 2020*; *Figure 3d* and *Figure 3—figure supplement 1*), we used the same decoding approach as described above, except that cross-validation was performed across trials corresponding to different stimulus conditions. Specifically, following the approach outlined in *Bernardi et al., 2020*, because there were three binary stimuli in our task (color, shape, and width), there were eight different trial conditions (*Figure 1b*). Therefore, for each task variable (we focused on color, shape, width, and the XOR between color and shape), there were six different ways of choosing two of the four conditions corresponding to each of the two possible stimuli for that task variable (e.g. color 1 vs. color 2). For example, when training a decoder to decode color, there were six different ways of choosing two conditions corresponding to color 1, and six different ways of choosing two conditions corresponding to color 2 (the remaining four conditions were then used for testing the decoder). Therefore, for each task variable, there were 6×6 = 36 different ways of creating training and testing sets that corresponded to different stimulus conditions. We then took the mean decoding accuracy we obtained across all 36 different training and testing sets.

## Measuring stimulus coding strength

In line with our mathematical analysis (Mathematical analysis of relevant and irrelevant stimulus coding in a linear network, and in particular *Equation S1*) and in line with previous studies (*Mante et al., 2013*; *Dubreuil et al., 2022*), to measure the strength of stimulus coding for relevant and irrelevant stimuli, we fitted the following linear regression model

$$\mathbf{r} = X\beta + \epsilon \tag{10}$$

where $\mathbf{r}$ is size $K{\times}N$ (where $K$ is the number of trials) and corresponds to the neural firing rates, $\mathbf{X}$ is size $K \times 3$ where the first column is all 1 (thus encoding the mean firing rate of each neuron across all conditions) and elements of the final two columns are either –0.5 or 0.5 depending upon whether trials correspond to XOR 1 or XOR 2 (relevant) conditions (column 2) or whether trials correspond to width 1 or width 2 (irrelevant) conditions (column 3). The coefficients to be fitted ($\beta$) is size $3 \times N$ and has the following structure $\beta = \left[ \boldsymbol{\mu}, \Delta\mathbf{r}_{\mathrm{rel}}, \Delta\mathbf{r}_{\mathrm{irrel}} \right]^{\top}$, where $\mu$ is the mean across all conditions for each neuron, $\Delta\mathbf{r}_{\mathrm{rel}}$ are the coefficients corresponding to relevant (XOR) conditions, and $\Delta\mathbf{r}_{\mathrm{irrel}}$ are the coefficients corresponding to irrelevant (width) conditions. Finally, $\epsilon$ is size $K{\times}N$ and contains the residuals. Note that calculating the mean difference in firing rate between the two relevant conditions and between the two irrelevant conditions would yield identical estimations of $\Delta\mathbf{r}_{\mathrm{rel}}$ and $\Delta\mathbf{r}_{\mathrm{irrel}}$ because our stimuli are binary. (We also fitted decoders to decode either relevant or irrelevant conditions and extracted their coefficients to instead obtain $\Delta\mathbf{r}_{\mathrm{rel}}$ and $\Delta\mathbf{r}_{\mathrm{irrel}}$ and obtained near-identical results.)

We then calculated the Euclidean norm of both $\Delta\mathbf{r}_{\mathrm{rel}}$ and $\Delta\mathbf{r}_{\mathrm{irrel}}$ normalized by the number of neurons ($\frac{\|\Delta\mathbf{r}_{\mathrm{rel}}\|_2}{\sqrt{N}}$ and $\frac{\|\Delta\mathbf{r}_{\mathrm{irrel}}\|_2}{\sqrt{N}}$) and the absolute value of the normalized dot product (overlap) between them ($\frac{|\Delta\mathbf{r}_{\mathrm{rel}}^{\top}\Delta\mathbf{r}_{\mathrm{irrel}}|}{\|\Delta\mathbf{r}_{\mathrm{rel}}\|_2\|\Delta\mathbf{r}_{\mathrm{irrel}}\|_2}$; *Figure 4e–i*). For our neural recordings (*Figure 4h*), we calculated these quantities

separately for 10 non-overlapping splits of the data. For our neural networks (*Figure 4e–g, i*), we calculated these quantities separately for 10 different networks.

## Measuring the magnitude of neural firing rates

We first calculated the firing rate for each condition and time point and then calculated the mean (across conditions and time points) Euclidean norm of firing rates appropriately normalized by the number of neurons: $\frac{\|\mathbf{r}(t)\|_2}{\sqrt{N}}$ (*Figure 5f* and *Figure 5—figure supplement 1d*). For our neural recordings (*Figure 5f*, far left), we calculated this separately for 10 non-overlapping splits of the data. For our neural networks (*Figure 5f*, middle left, middle right, and far right, and *Figure 5—figure supplement 1d*), we calculated this separately for 10 different networks. For our optimized neural networks, to emulate alternative tasks where different combinations of color, shape, and width are relevant, we also performed the same analysis when shuffling all 6 input weights (after learning) in all 14 possible ways (*Figure 5—figure supplement 1d*, blue lines, 'shuffle inputs'). These 14 possible shuffles excluded the original setup of input weights, any re-ordering of inputs within a single input channel (since any re-ordering within a single input channel would be identical up to a re-labeling), and any re-ordering between the shape and width input channels (since any re-ordering within the shape and width input channels would also be identical up to a re-labeling).

### Statistics

For decoding analyses, we used non-parametric permutation tests to calculate statistical significance. We used 100 different random shuffles of condition labels across trials to generate a null distribution for decoding accuracy. We plotted chance level decoding (*Figure 3a and b*, *Figure 2—figure supplement 1*, *Figure 3—figure supplements 1–3*, and *Figure 5—figure supplement 1a and b*) by combining both early and late learning null distributions.

To calculate significant differences in decoding accuracy over learning (*Figure 3a and b*, *Figure 2—figure supplement 1*, *Figure 3—figure supplements 1–3*, and *Figure 5—figure supplement 1a and b*), our test statistic was the difference in decoding accuracy between early and late learning, and our null distribution was the difference in decoding accuracy between early and late learning for the 100 different shuffles of condition labels (see above). We calculated two-tailed p-values for all tests. Additionally, to control for time-related cluster-based errors, we also added a cluster-based permutation correction (*Maris and Oostenveld, 2007*).

For all other tests, we used a two-sided Mann–Whitney U test (*Figures 4h, i and 5f*, and *Figure 5—figure supplement 1d*).

### Randomization

No new experimental data was gathered for this paper. Previously collected experimental data contained no groups. Trial types were randomly determined by a computer program.

### Blinding

As data collection had been performed well before the development of our models and our corresponding analyses were performed, it was effectively blind to the purposes of our study. Data analysis was not performed blind to the conditions of the experiments. For previously collected experimental data, there was no blinding as there was no group allocation.

## Acknowledgements

This work was funded by the Wellcome Trust (Investigator Award in Science 212262/Z/18/Z to ML, Sir Henry Wellcome Postdoctoral Fellowship 215909/Z/19/Z to JS, and award 101092/Z/13/Z to MK, MK, and JD), the Human Frontier Science Programme (Research Grant RGP0044/2018 to ML), the Medical Research Council UK Program (MC_UU_00030/7 to MK, MK, and JD), the Biotechnology and Biological Sciences Research Council (award BB/M010732/1 to MGS), a Gates Cambridge scholarship (to KTJ), a Clarenden scholarship and Saven European scholarship (to MW), and the James S McDonnell Foundation (award 220020405 to MGS). For the purpose of open access, the authors have applied a CC-BY public copyright licence to any author accepted manuscript version arising from this

submission. We thank Christopher Summerfield for useful feedback and detailed comments on the manuscript

## Additional information

### Funding

| Funder | Grant reference number | Author |
|---|---|---|
| Wellcome Trust | 10.35802/215909 | Jake Patrick Stroud |
| Wellcome Trust | 10.35802/212262 | Mate Lengyel |
| Wellcome Trust | 10.35802/101092 | Makoto Kusunoki<br>Mikiko Kadohisa<br>John Duncan |
| Human Frontier Science Program | RGP0044/2018 | Mate Lengyel |
| Medical Research Council | MC_UU_00030/7 | Makoto Kusunoki<br>Mikiko Kadohisa<br>John Duncan |
| Biotechnology and Biological Sciences Research Council | BB/M010732/1 | Mark G Stokes |
| Gates Cambridge Trust | | Kristopher Torp Jensen |
| Clarendon Fund | | Michal Wojcik |
| James S. McDonnell Foundation | 10.37717/220020405 | Mark G Stokes |

The funders had no role in study design, data collection and interpretation, or the decision to submit the work for publication. For the purpose of Open Access, the authors have applied a CC BY public copyright license to any Author Accepted Manuscript version arising from this submission.

### Author contributions

Jake Patrick Stroud, Conceptualization, Resources, Formal analysis, Funding acquisition, Validation, Investigation, Visualization, Methodology, Writing – original draft, Project administration, Writing – review and editing; Michal Wojcik, Data curation, Formal analysis, Methodology, Writing – original draft, Writing – review and editing; Kristopher Torp Jensen, Conceptualization, Formal analysis, Investigation, Methodology, Writing – original draft, Writing – review and editing; Makoto Kusunoki, Mikiko Kadohisa, Data curation, Writing – review and editing; Mark J Buckley, Data curation, Funding acquisition, Writing – review and editing; John Duncan, Conceptualization, Supervision, Funding acquisition, Methodology, Writing – original draft, Writing – review and editing; Mark G Stokes, Conceptualization, Data curation, Investigation, Methodology, Writing – review and editing; Mate Lengyel, Conceptualization, Resources, Formal analysis, Supervision, Investigation, Visualization, Methodology, Writing – original draft, Project administration, Writing – review and editing

### Author ORCIDs

Jake Patrick Stroud ⓘ https://orcid.org/0000-0002-4263-5755
Kristopher Torp Jensen ⓘ https://orcid.org/0000-0001-9242-5572
Makoto Kusunoki ⓘ https://orcid.org/0000-0002-5381-8506
Mark J Buckley ⓘ https://orcid.org/0000-0001-7455-8486

Reviewer #1 (Public Review): https://doi.org/10.7554/eLife.94961.2.sa1
Reviewer #2 (Public Review): https://doi.org/10.7554/eLife.94961.2.sa2
Reviewer #3 (Public Review): https://doi.org/10.7554/eLife.94961.2.sa3
Author response https://doi.org/10.7554/eLife.94961.2.sa4

## Additional files

**Supplementary files**
MDAR checklist

**Data availability**
All code was custom written in Python using NumPy, SciPy, Matplotlib, Scikit-learn, and Tensorflow. All code is available in GitHub (copy archived at *Stroud, 2023*). All neural recordings have been deposited at Dryad.

The following previously published dataset was used:

| Author(s) | Year | Dataset title | Dataset URL | Database and Identifier |
|---|---|---|---|---|
| Wojic M, Stroud J, Wasmuht D, Kusunoki M, Kadohisa M, Buckley M, Myers N, Hunt L, Duncan J, Stokes M | 2024 | Electrophysiological recordings of prefrontal activity over learning in non-human primates | https://doi.org/10.5061/dryad.c2fqz61kb | Dryad Digital Repository, 10.5061/dryad.c2fqz61kb |

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

# Appendix 1

In this supplementary appendix, we derive the optimal linear decoder for a linear network that receives both relevant and irrelevant inputs (Problem definition and The optimal linear decoder). We derive how both the performance of the optimal decoder (The performance of the optimal linear decoder) and the metabolic cost of the network (Metabolic cost) depend on noise and the strength of relevant and irrelevant inputs. We also derive how the optimal setting of the relevant and irrelevant inputs, when jointly optimizing for both performance and a metabolic cost, depend on noise and the strength of metabolic cost (Qualitative predictions about optimal parameters). Finally, we also study how noise and the strength of metabolic cost affect the curvature of the loss function around the optimum (Curvature of the loss function around the optimum).

## Mathematical analysis of relevant and irrelevant stimulus coding in a linear network

Even though in the main text we simulate and numerically analyze neural networks with nonlinear temporal dynamics that solve an XOR task, for analytical tractability, our mathematical analyses below are for a model in which responses to different inputs combine linearly. Our analysis is agnostic as to whether responses come about by simple, instantaneous feedforward or temporally extended recurrent interactions, and is only concerned with the phenomenological mapping between stimuli and the resulting response distributions. As such, we cannot distinguish between dynamically and statically irrelevant stimuli, and so we only include a single relevant and single (statically) irrelevant stimulus in our analysis. Nevertheless, as we show in the main text (*Figure 4*), some of the key insights from our analysis of such simplified systems generalize to the original setting.

### Problem definition

We consider the responses, $\mathbf{r}$, of a neural population of $N$ neurons to a stimulus that has a 'relevant' feature, $c$, which we assume to be binary with values $c_1$ and $c_2$ (occurring with uniform frequency), and to a (scalar) irrelevant feature, $\epsilon$ that is continuous (for convenience). We further assume that the responses are determined by a linear interaction of the relevant and irrelevant features and also subject to unspecific neural noise:

$$\mathbf{r} = \boldsymbol{\mu} + \delta_{c,c_1}\frac{\Delta\mathbf{r}_{\text{rel}}}{2} - \delta_{c,c_2}\frac{\Delta\mathbf{r}_{\text{rel}}}{2} + \epsilon\frac{\Delta\mathbf{r}_{\text{irrel}}}{2} + \boldsymbol{\eta} \tag{S1}$$

where $\mu$ is the grand average response, averaging across both the relevant and the irrelevant feature, $\Delta\mathbf{r}_{\text{rel}}$ is the relevant tuning of the population, $\Delta\mathbf{r}_{\text{irrel}}$ is the irrelevant tuning, $\epsilon$ is (without loss of generality) zero mean and unit variance variability in the irrelevant feature, and $\boldsymbol{\eta} \sim \mathcal{N}\left(0, \sigma^2\boldsymbol{I}\right)$ is other neural noise.

In this note, we study how this neural population can optimize the decodability of $c$ while also balancing metabolic cost (defined below). Specifically, we will regard $\boldsymbol{\mu}$ and $\sigma^2$ as givens (constraints; one could also consider $\boldsymbol{\mu}$ as an optimizable parameter, and its optimal value would simply be $\mathbf{0}$) and ask how the population should 'choose' $\Delta\mathbf{r}_{\text{rel}}$ and $\Delta\mathbf{r}_{\text{irrel}}$ for this trade-off (or some summary statistics thereof, see below).

### The optimal linear decoder

We first study in general how information can be decoded from the population. The statistically optimal decoder, decoding $c$ from $\mathbf{r}$, is the maximum likelihood decoder (note that we have assumed $c_1$ and $c_2$ to occur with equal probabilities, see above). To make the derivations tractable, from here on we assume that $\epsilon$ is specifically Gaussian distributed, i.e., $\epsilon \sim \mathcal{N}\left(0, 1\right)$, unless otherwise noted. In this case, the distribution of responses conditioned on the relevant stimulus feature becomes (equivariate) Gaussian, with some effective noise covariance $\boldsymbol{\Sigma}$ (to be determined later, *Equation S13*):

$$r|c_1 \sim \mathcal{N}\left(\boldsymbol{\mu} + \frac{\Delta\mathbf{r}_{\text{rel}}}{2}, \boldsymbol{\Sigma}\right) \tag{S2}$$

$$r|c_2 \sim \mathcal{N}\left(\boldsymbol{\mu} - \frac{\Delta \mathbf{r}_{\mathrm{rel}}}{2}, \boldsymbol{\Sigma}\right) \tag{S3}$$

Thus, the log odds ratio is

$$z = ln\frac{\mathcal{P}\left(\mathbf{r}|c_1\right)}{\mathcal{P}\left(\mathbf{r}|c_2\right)} = \Delta \mathbf{r}_{\mathrm{rel}}^{\top}\boldsymbol{\Sigma}^{-1}\mathbf{r} - \Delta \mathbf{r}_{\mathrm{rel}}^{\top}\boldsymbol{\Sigma}^{-1}\boldsymbol{\mu} \tag{S4}$$

and the optimal decoder responds '1' if

$$z > 0 \tag{S5}$$

Thus, in this case, the optimal decoder is linear in $\mathbf{r}$, or conversely, a linear decoder (with the correct coefficients) is optimal.

## The performance of the optimal linear decoder

We now turn to the question of how the parameters of the neural population affect its decodability, i.e., the performance of the optimal linear decoder. (We still assume that $\epsilon$, and thus $\mathbf{r}|c_{1/2}$, is Gaussian distributed.)

As we saw, the optimal decoder can be described by a simple thresholding of the log odds (*Equation S5*). The log odds, $z$, itself is also Gaussian distributed conditioned on the relevant feature, because $\mathbf{r}$ is Gaussian distributed (*Equations S2 and S3*) and $z$ is a linear function of it (*Equation S4*):

$$z|c_1 \sim \mathcal{N}\left(\mu_z, \sigma_z^2\right) \tag{S6}$$

and, due to symmetry,

$$z|c_2 \sim \mathcal{N}\left(-\mu_z, \sigma_z^2\right) \tag{S7}$$

with

$$\mu_z = \mathbb{E}\left[z|c_1\right] = -\mathbb{E}\left[z|c_2\right] = \frac{1}{2}\Delta \mathbf{r}_{\mathrm{rel}}^{\top}\boldsymbol{\Sigma}^{-1}\Delta \mathbf{r}_{\mathrm{rel}} \tag{S8}$$

and

$$\sigma_z^2 = \mathbb{V}\left[z|c_1\right] = \mathbb{V}\left[z|c_2\right] = \Delta \mathbf{r}_{\mathrm{rel}}^{\top}\boldsymbol{\Sigma}^{-1}\Delta \mathbf{r}_{\mathrm{rel}} = 2\mu_z \tag{S9}$$

The probability of correct decoding for $c_1$ (and by symmetry, also for $c_2$, and thus also after averaging over $c$) is given by

$$\Pi = \mathcal{P}\left(\text{respond } 1|c_1\right) = \mathcal{P}\left(\text{respond } 2|c_2\right) \tag{S10}$$

$$= \int_0^{\infty} \mathcal{N}\left(z; \mu_z, \sigma_z^2\right) dz = \Psi\left(\frac{\mu_z}{\sigma_z}\right) = \Psi\left(\sqrt{\frac{\mu_z}{2}}\right) \tag{S11}$$

Therefore, the performance of the optimal linear decoder scales monotonically with $\mu_z$. To gain further insight into what this means in our particular setting, let us now express the effective noise covariance matrix $\boldsymbol{\Sigma}$ with the parameters of the problem definition:

$$\boldsymbol{\Sigma} = \mathbb{C}\left[\mathbf{r}|c_1\right] = \mathbb{C}\left[\mathbf{r}|c_2\right] \tag{S12}$$

$$= \sigma^2 \mathbf{I} + \frac{\Delta \mathbf{r}_{\mathrm{irrel}}\Delta \mathbf{r}_{\mathrm{irrel}}^{\top}}{4} \tag{S13}$$

and its inverse (expressed using the Sherman–Morrison formula):

$$\boldsymbol{\Sigma}^{-1} = \frac{1}{\sigma^2}\left[\mathbf{I} - \frac{\Delta \mathbf{r}_{\mathrm{irrel}}\Delta \mathbf{r}_{\mathrm{irrel}}^{\top}}{4\sigma^2 + \Delta \mathbf{r}_{\mathrm{irrel}}^{\top}\Delta \mathbf{r}_{\mathrm{irrel}}}\right] \tag{S14}$$

Substituting *Equation S14* into the formula for $\mu_z$ (*Equation S8*), we obtain:

$$\mu_z = \frac{1}{2\sigma^2}\Delta\mathbf{r}_{\mathrm{rel}}^\top\Delta\mathbf{r}_{\mathrm{rel}} - \frac{1}{2\sigma^2}\frac{\left(\Delta\mathbf{r}_{\mathrm{rel}}^\top\Delta\mathbf{r}_{\mathrm{irrel}}\right)\left(\Delta\mathbf{r}_{\mathrm{irrel}}^\top\Delta\mathbf{r}_{\mathrm{rel}}\right)}{4\sigma^2 + \Delta\mathbf{r}_{\mathrm{irrel}}^\top\Delta\mathbf{r}_{\mathrm{irrel}}} \tag{S15}$$

$$= \frac{1}{2\sigma^2}\|\Delta\mathbf{r}_{\mathrm{rel}}\|^2\left[1 - \frac{\|\Delta\mathbf{r}_{\mathrm{irrel}}\|^2}{4\sigma^2 + \|\Delta\mathbf{r}_{\mathrm{irrel}}\|^2}\left(\frac{\Delta\mathbf{r}_{\mathrm{rel}}^\top\Delta\mathbf{r}_{\mathrm{irrel}}}{\|\Delta\mathbf{r}_{\mathrm{rel}}\|\,\|\Delta\mathbf{r}_{\mathrm{irrel}}\|}\right)^2\right] \tag{S16}$$

Thus, $\mu_z$ can be simply expressed as

$$\mu_z = \frac{1}{2\sigma^2}\gamma^2\left(1 - \frac{\alpha^2}{4\sigma^2 + \alpha^2}\beta^2\right) \tag{S17}$$

where

$$\gamma = \|\Delta\mathbf{r}_{\mathrm{rel}}\| \text{ is the magnitude of tuning to the relevant feature} \tag{S18}$$

$$\alpha = \|\Delta\mathbf{r}_{\mathrm{irrel}}\| \text{ is the magnitude of tuning to the irrelevant feature} \tag{S19}$$

$$\text{and } \beta = \frac{\Delta\mathbf{r}_{\mathrm{rel}}^\top\Delta\mathbf{r}_{\mathrm{irrel}}}{\gamma\alpha} \text{ is the overlap of irrelevant with relevant tuning} \tag{S20}$$

which reveals that the effects of noise, relevant tuning, and irrelevant tuning factorize (corresponding to the first, second, and third terms, respectively), and that – intuitively – $\mu_z$ and thus performance increases with $\gamma$ and decreases with $\alpha$, $\beta$, and $\sigma^2$ (where we always consider the latter a constraint and thus fixed, see the problem definition).

Note that in the small noise limit, $\sigma^2 \ll \alpha^2$:

$$\mu_z = \frac{1}{2\sigma^2}\gamma^2\left(1 - \beta^2\right) \tag{S21}$$

showing that performance in this case can only be increased by decreasing $\beta$ (or, trivially, by increasing $\gamma$), but not by decreasing $\alpha$. In contrast, when the small noise limit does not hold, the original *Equation S17* applies, and so performance can be increased by decreasing either $\alpha$ or $\beta$ (or, again, by increasing $\gamma$).

## Metabolic cost

Following previous work (*Wójcik, 2023*; *Rigotti et al., 2013*; *Yang et al., 2019*; *Wang et al., 2018*; *Silver et al., 2016*; *Jensen et al., 2023*), we define the metabolic cost to be the average sum of squared neural responses (where the averaging is over realizations of the relevant and irrelevant features as well as neural noise):

$$\omega^2 = \mathbb{E}\left[\|\mathbf{r}\|^2\right] \tag{S22}$$

which, by making the averaging over relevant features explicit, can be rewritten as

$$\omega^2 = \frac{\mathbb{E}\left[\|\mathbf{r}\|^2\,|c_1\right] + \mathbb{E}\left[\|\mathbf{r}\|^2\,|c_2\right]}{2} \tag{S23}$$

where the metabolic cost for $c_1$ is

$$\mathbb{E}\left[\|\mathbf{r}\|^2\,|c_1\right] = \mathbb{E}\left[\mathbf{r}|c_1\right]^\top\mathbb{E}\left[\mathbf{r}|c_1\right] + \mathrm{Tr}\left(\mathbb{C}\left[\mathbf{r}|c_1\right]\right) \tag{S24}$$

$$= \left(\boldsymbol{\mu} + \frac{\Delta\mathbf{r}_{\mathrm{rel}}}{2}\right)^\top\left(\boldsymbol{\mu} + \frac{\Delta\mathbf{r}_{\mathrm{rel}}}{2}\right) + \mathrm{Tr}\left(\boldsymbol{\Sigma}\right) \tag{S25}$$

$$= \|\boldsymbol{\mu}\|^2 + \frac{\gamma^2}{4} + \boldsymbol{\mu}^\top \Delta \mathbf{r}_{\text{rel}} + \frac{\alpha^2}{4} + N\sigma^2 \tag{S26}$$

and, again by symmetry,

$$\mathbb{E}\left[\|\mathbf{r}\|^2 \,|c_2\right] = \|\boldsymbol{\mu}\|^2 + \frac{\gamma^2}{4} - \boldsymbol{\mu}^\top \Delta \mathbf{r}_{\text{rel}} + \frac{\alpha^2}{4} + N\sigma^2 \tag{S27}$$

and so, by substituting *Equations S26 and S27* into *Equation S23*, we get:

$$\omega^2 = \|\boldsymbol{\mu}\|^2 + N\sigma^2 + \frac{\gamma^2}{4} + \frac{\alpha^2}{4} \tag{S28}$$

This result is also interesting, because it shows that the metabolic cost factorizes into four additive terms, each of which scales monotonically with a separate parameter: the average magnitude of responses, neural noise, the magnitude of relevant tuning, and the magnitude of irrelevant coding. It is also interesting to note what the metabolic cost does *not* depend on: the overlap between relevant and irrelevant tuning, $\beta$.

The first two terms in *Equation S28* are assumed to be fixed (see problem definition, Problem definition), so we will not consider them further. The last two terms in *Equation S28* increase with $\gamma$ and $\alpha$, respectively. As we want the metabolic cost to be small, this prefers small $\gamma$ and $\alpha$.

Note that, unlike for deriving the optimal linear decoder (The optimal linear decoder) and its performance (The performance of the optimal linear decoder), for which we needed to assume that $\epsilon$ is normally distributed, no such assumption needed to be made for computing the metabolic cost (*Equation S28*). Thus, for the metabolic cost, the same result is obtained for example when the irrelevant feature is binary, such that $\epsilon = \pm 1$ with equal probability (without loss of generality; note that, in this case, it is still true that $\mathbb{E}\left[\epsilon\right] = 0$ and $\mathbb{V}\left[\epsilon\right] = 1$, which is all that we assumed in the derivation above (see *Equation S13*)), i.e., when variability in the irrelevant feature simply changes the responses by $\pm \Delta \mathbf{r}_{\text{irrel}}$.

## Qualitative predictions about optimal parameters

In general, the optimal setting of optimizable parameters, $\Delta \mathbf{r}_{\text{rel}}$ and $\Delta \mathbf{r}_{\text{irrel}}$ (Problem definition), depends on the overall objective function, which will usually be a sum of performance (which we want to be high) and (negative) metabolic cost (which we want to be low), with some suitable Lagrange multiplier, $\lambda$, controlling the trade-off between the two terms:

$$\mathcal{L}\left(\gamma, \alpha, \beta\right) = \underbrace{\Pi}_{\text{Eqs. S11 and S17}} - \lambda \underbrace{\omega^2}_{\text{Eq. S28}} \tag{S29}$$

We note that the objective function also depends on the constraint parameters, $\boldsymbol{\mu}$ and $\sigma$ (Problem definition). The effect of $\mu$ is trivial: it only affects the metabolic cost, not performance, and it does so as a simple additive term (*Equation S28*), so it does not affect the optimal values of the other parameters. (This remains true even if $\boldsymbol{\mu}$ is optimizable, in which case it is also obvious from *Equation S28* that its optimal value is **0**.) The effect of the other constraint, $\sigma$, is more nuanced, so we will separately consider two different regimes for it: small noise ($0 \leq \sigma \leq \sigma_{\text{crit}}$) and large noise ($\sigma_{\text{crit}} < \sigma$) with the constant $\sigma_{\text{crit}}$ defined later.

We also note that both performance (*Equation S17*) and metabolic cost (*Equation S28*) only depend on the optimizable parameters, $\Delta \mathbf{r}_{\text{rel}}$ and $\Delta \mathbf{r}_{\text{irrel}}$, through their summary statistics, $\gamma$, $\alpha$, and (for performance) $\beta$ (*Equations S18–S20*). In general, the dependence on the latter two parameters is straightforward: performance decreases in both $\alpha$ and $\beta$ (so that the last term in *Equation S17* achieves its maximal possible value 1 when either parameter is 0), while the metabolic cost increases with $\alpha$. This means that their optimal values will be at 0. However, we recall that in the small noise limit, only decreasing $\beta$ can improve performance (*Equation S21*), while otherwise, decreasing $\alpha$ can also make a contribution to it. At the same time, the metabolic cost only depends on $\alpha$ but not on $\beta$. Thus to summarize, in the small noise limit, there is 'pressure' on both $\beta$ (from performance) and $\alpha$ (from the metabolic cost) to be small. In other cases, the pressure on $\alpha$ comes from both performance and metabolic cost, while $\beta$ only matters for performance, so we expect $\alpha$ to be more aggressively minimized by optimization, and perhaps $\beta$ not so much (which we find to be the case

for our optimized recurrent neural networks; *Figure 4f and g*; see also Curvature of the loss function around the optimum).

The overall effect of $\gamma$ is slightly less trivial. *Equation S28* shows that the metabolic cost depends on it quadratically, i.e., it should be small. However, as we saw earlier (*Equation S17*), decoding performance monotonically grows with it. Nevertheless, this dependence of decoding performance on $\gamma$ is nonlinear (through the standard normal c.d.f., *Equation S11*), such that it is effectively linear in $\gamma$ for small values, but saturates for large values of $\gamma$. (This is because $\Psi(x)$ is linear in the $x{\to}0$ limit, and the argument of $\Psi(\cdot)$ for computing the performance is linear in the square root of $\mu_z$ (*Equation S11*), which in turn is quadratic in $\gamma$ (*Equation S17*), so the argument of $\Psi(\cdot)$ is linear in $\gamma$.) This means that for small values of $\gamma$ the linear performance will dominate over the quadratic metabolic cost, but eventually, for large values, the quadratic metabolic term is guaranteed to dominate over the saturating performance, thus effectively limiting the magnitude of relevant tuning to some finite value. Note, however, that this argument does not yet reveal what happens to $\gamma$ depending on the noise regime, and where the transition between these two regimes happens.

Fortunately, it is possible to analytically derive $\gamma_*$, the optimal value of $\gamma$, as a function of $\sigma$. For this, we assume that $\alpha$ and $\beta$ already take their optimal values (without loss of generality, as we are interested in jointly optimizing all relevant parameters), such that the last term in *Equation S17* is simply 1 (see above). In this case, the terms in the overall objective function (*Equation S29*) that depend on the two remaining parameters of interest ($\gamma$ and $\sigma$) are simply:

$$\mathcal{L} = \Psi\left(\frac{\gamma}{2\sigma}\right) - \frac{\lambda}{4}\gamma^2 - \lambda N\sigma^2 + \ldots \tag{S30}$$

The optimal $\gamma$ can be simply defined implicitly as a function of $\sigma$ as the solution to the following equation:

$$0 = \frac{\partial \mathcal{L}}{\partial \gamma}\left(\gamma_*, \sigma\right) \tag{S31}$$

We now substitute (the partial derivative of) *Equation S30* into *Equation S31* to obtain:

$$0 = \frac{1}{2\sigma}\mathcal{N}\left(\frac{\gamma_*}{2\sigma}\right) - \frac{\lambda}{2}\gamma_* \tag{S32}$$

Note that this yields a solution for any $\sigma, \lambda > 0$ since the line $\frac{\lambda}{2}\gamma_*$ intersects the pdf of the Normal distribution for some $\gamma_* \geq 0$. Re-arranging *Equation S32* gives us:

$$\gamma_* e^{\frac{\gamma_*^2}{8\sigma^2}} = \frac{1}{\sqrt{2\pi}\sigma\lambda} \tag{S33}$$

This equation has a solution in terms of the Lambert $W$ function (defined by its inverse as $W^{-1}(x) = xe^x$)

$$\gamma_* = 2\sigma\sqrt{W\left(\frac{1}{8\pi\lambda^2\sigma^4}\right)} \tag{S34}$$

where we only take the positive solution since $\gamma$ can only be positive.

We can find the maximum of this function by taking the derivative of *Equation S34* and setting it equal to 0. This gives:

$$W\left(\frac{1}{8\pi\lambda^2\sigma_{\text{crit}}^4}\right) = 1 \tag{S35}$$

Using the fact that $W(e) = 1$, we obtain:

$$\sigma_{\text{crit}} = \frac{1}{\left(8\pi\lambda^2 e\right)^{1/4}} \tag{S36}$$

Substituting this into **Equation S34** gives:

$$\gamma_{\text{crit}} = 2\sigma_{\text{crit}} \tag{S37}$$

These results are shown in **Appendix 1—figure 1**, together with a numerical confirmation (see also **Figure 4d**).

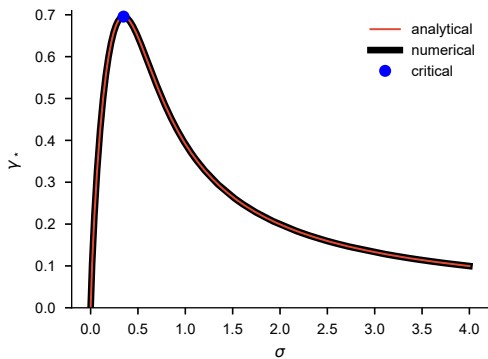

**Appendix 1—figure 1.** Plot of $\gamma*$ as a function of $\sigma$ obtained by numerically optimizing **Equation S30** (black), or using the analytical expression in **Equation S34** (red). Blue dot shows (analytically computed) critical values where $\gamma*$ has a maximum (**Equations S36 and S37**). We used $\lambda=1$ for these results.

The qualitative dependence of $\gamma_*$ on $\sigma$ is straightforward to interpret and intuitive. In the *small* noise regime, $0 \leq \sigma \leq \sigma_{\text{crit}}$, $\gamma$ needs to grow with $\sigma$ so that the separation of the response distributions conditioned on $c_1$ and $c_2$ remains large enough to guarantee high performance. This growth starts at zero because for zero noise any infinitesimal separation between the mean responses for $c = c_1$ and $c_2$ makes for perfect performance. Thus, $\gamma$ remains small, and so – as we saw above – the performance term in the objective function dominates over the metabolic cost term. This means that the value of $\gamma$ that optimizes the full objective function can be understood from just this performance-based perspective. However, in the *large* noise regime, $\sigma_{\text{crit}} < \sigma$, $\gamma$ would need to be so large to guarantee high performance that it would reach a regime in which – as we saw above – the metabolic cost dominates. Thus the optimal $\gamma$ is increasingly influenced by the metabolic cost, and thus decreases with $\sigma$.

Finally, we derive classification performance with optimized parameters as a function of $\sigma$ by substituting **Equation S34** into **Equations S11 and S17** and obtain:

$$\Pi = \Psi \left( \sqrt{W \left( \frac{1}{8\pi\lambda^2\sigma^4} \right)} \right) \tag{S38}$$

which shows the intuitive result that performance monotonically decreases with $\sigma$ and drops to chance level for $\sigma \to \infty$ (**Appendix 1—figure 2**, red). This is because the argument of $W$ monotonically decreases with $\sigma$ from infinity to zero, while both $W(x)$ and $\sqrt{x}$ monotonically increase (without bounds) with $x$ from zero at $x=0$, and finally $\Psi(x)$ also monotonically increases with $x$ but from $\frac{1}{2}$ at $x=0$ and to an asymptotic bound of 1.

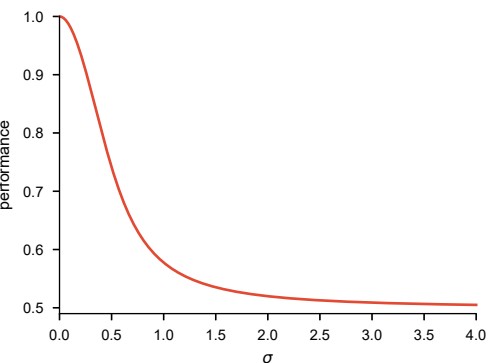

**Appendix 1—figure 2.** Plot of $\mathcal{P}\left(\text{correct}\right)$ as a function of $\sigma$ when using $\gamma = \gamma*$ that is optimized as in *Appendix 1—figure 1* (*Equation S38*). We used $\lambda=1$ for these results. All other parameters were optimized in all cases.

## Curvature of the loss function around the optimum

The curvature of the loss landscape around the optimum is important for determining how tightly constrained the parameters will be in the presence of noisy gradient updates due to the finite batch sizes used for stochastic gradient descent. Before we proceed, we first repeat here the main equations regarding the loss $\mathcal{L}$:

$$\mathcal{L} = \Pi - \lambda\omega^2 \tag{S39}$$

$$\Pi = \Psi\left(\sqrt{\frac{\mu_z}{2}}\right) \tag{S40}$$

$$\mu_z = \frac{1}{2\sigma^2}\gamma^2\left(1 - \frac{\alpha^2}{4\sigma^2 + \alpha^2}\beta^2\right) \tag{S41}$$

$$\omega^2 = \|\boldsymbol{\mu}\|^2 + N\sigma^2 + \frac{\gamma^2}{4} + \frac{\alpha^2}{4} \tag{S42}$$

### Curvature of the loss function with respect to $\alpha$

In this section, we are interested in the first non-zero term of the Taylor expansion of the loss function $\mathcal{L}$ with respect to $\alpha$ – the magnitude of the irrelevant input – around the optimum at $\alpha = \beta = 0$. It turns out that the first non-zero term results from the second derivative of the metabolic cost term in the loss function since the other terms always contain at least an $\alpha$ or $\beta$ (which are 0 at the optimum). We therefore find that:

$$\frac{\partial^2\mathcal{L}}{\partial\alpha^2}|_{\alpha=0,\beta=0} = \frac{\partial^2\Pi}{\partial\alpha^2}|_{\alpha=0,\beta=0} - \lambda\frac{\partial^2\omega^2}{\partial\alpha^2}|_{\alpha=0,\beta=0} \tag{S43}$$

$$= 0 - \lambda\frac{\partial^2}{\partial\alpha^2}\left[\|\boldsymbol{\mu}\|^2 + N\sigma^2 + \frac{\gamma^2}{4} + \frac{\alpha^2}{4}\right] \tag{S44}$$

$$= -\frac{1}{2}\lambda. \tag{S45}$$

From this result, we see that the magnitude of the curvature of the loss decreases as a function of $\lambda$. In other words, the loss landscape as a function of the irrelevant input $\alpha$ becomes steeper with increasing regularization. This is why we observed larger values of $\alpha = \|\Delta\mathbf{r}_{\text{irrel}}\|$ (*Equation S18*) in *Figure 4f* with decreasing $\lambda$.

## Curvature of the loss function with respect to $\beta$

We now turn our attention to the first non-zero term of the Taylor expansion of the loss function $\mathcal{L}$ with respect to $\beta$ – the overlap of irrelevant with relevant tuning – around the optimum at $\alpha = \beta = 0$. It turns out that the first non-zero term results from a fourth derivative of the performance term $\Pi$ with respect to both $\alpha$ and $\beta$:

$$\frac{\partial^4 \mathcal{L}}{\partial \alpha^2 \beta^2}|_{\alpha=0,\beta=0} = \frac{\partial^4 \Pi}{\partial \alpha^2 \beta^2}|_{\alpha=0,\beta=0} - \lambda \frac{\partial^4 \omega^2}{\partial \alpha^2 \beta^2}|_{\alpha=0,\beta=0} \tag{S46}$$

$$= \frac{\partial^4 \Pi}{\partial \alpha^2 \beta^2}|_{\alpha=0,\beta=0} - \lambda \frac{\partial^4}{\partial \alpha^2 \beta^2} \left[ \|\boldsymbol{\mu}\|^2 + N\sigma^2 + \frac{\gamma^2}{4} + \frac{\alpha^2}{4} \right]|_{\alpha=0,\beta=0} \tag{S47}$$

$$= \frac{\partial^4 \Pi}{\partial \alpha^2 \beta^2}|_{\alpha=0,\beta=0} - 0. \tag{S48}$$

$$= \frac{\partial^4 \Pi}{\partial \alpha^2 \beta^2}|_{\alpha=0,\beta=0} \tag{S49}$$

To evaluate this term we note that:

$$\frac{\partial^2 \Pi}{\partial \beta^2} = \frac{\partial^2 \Pi}{\partial \mu_z^2} \frac{\partial \mu_z}{\partial \beta} + \frac{\partial \Pi}{\partial \mu_z} \frac{\partial^2 \mu_z}{\partial \beta^2} \tag{S50}$$

$$= \frac{\partial^2 \Pi}{\partial \mu_z^2} \left( -\frac{\gamma^2}{2\sigma^2} \frac{2\alpha^2 \beta}{4\sigma^2 + \alpha^2} \right) + \frac{\partial \Pi}{\partial \mu_z} \left( -\frac{\gamma^2}{2\sigma^2} \frac{2\alpha^2}{4\sigma^2 + \alpha^2} \right) \tag{S51}$$

Therefore,

$$\frac{\partial^4 \Pi}{\partial \alpha^2 \beta^2}|_{\alpha=0,\beta=0} = \frac{\gamma^2}{2\sigma^2} \frac{\partial^2}{\partial \alpha^2} \left( \frac{\partial^2 \Pi}{\partial \mu_z^2} \left( -\frac{2\alpha^2 \beta}{4\sigma^2 + \alpha^2} \right) + \frac{\partial \Pi}{\partial \mu_z} \left( -\frac{2\alpha^2}{4\sigma^2 + \alpha^2} \right) \right)|_{\alpha=0,\beta=0} \tag{S52}$$

$$= \frac{\beta \gamma^2}{2\sigma^2} \frac{\partial^2}{\partial \alpha^2} \left( \frac{\partial^2 \Pi}{\partial \mu_z^2} \left( -\frac{2\alpha^2}{4\sigma^2 + \alpha^2} \right) \right) + \frac{\gamma^2}{2\sigma^2} \frac{\partial^2}{\partial \alpha^2} \left( \frac{\partial \Pi}{\partial \mu_z} \left( -\frac{2\alpha^2}{4\sigma^2 + \alpha^2} \right) \right)|_{\alpha=0,\beta=0} \tag{S53}$$

$$= \frac{\gamma^2}{2\sigma^2} \frac{\partial^2}{\partial \alpha^2} \left( \frac{\partial \Pi}{\partial \mu_z} \left( -\frac{2\alpha^2}{4\sigma^2 + \alpha^2} \right) \right)|_{\alpha=0,\beta=0} \tag{S54}$$

Now, noting that $\frac{\partial}{\partial \alpha} = \frac{\partial}{\partial \mu_z} \frac{\partial \mu_z}{\partial \alpha}$, we obtain:

$$\frac{\partial^2}{\partial \alpha^2} \left( \frac{\partial \Pi}{\partial \mu_z} \left( -\frac{2\alpha^2}{4\sigma^2 + \alpha^2} \right) \right)|_{\alpha=0,\beta=0} = -\frac{2\alpha^2}{4\sigma^2 + \alpha^2} \left( \frac{\partial^2 \Pi}{\partial \mu_z^2} \frac{\partial^2 \mu_z}{\partial \alpha^2} + \frac{\partial^3 \Pi}{\partial \mu_z^3} \left( \frac{\partial^2 \mu_z}{\partial \alpha} \right)^2 \right)$$
$$+ \frac{\partial \Pi}{\partial \mu_z} \frac{\partial^2}{\partial \alpha^2} \left( -\frac{2\alpha^2}{4\sigma^2 + \alpha^2} \right)|_{\alpha=0,\beta=0} \tag{S55}$$

$$= \frac{\partial \Pi}{\partial \mu_z} \frac{\partial^2}{\partial \alpha^2} \left( -\frac{2\alpha^2}{4\sigma^2 + \alpha^2} \right)|_{\alpha=0,\beta=0} \tag{S56}$$

$$= \frac{\partial \Pi}{\partial \mu_z} \frac{\partial}{\partial \alpha} \left( -\frac{4\alpha}{4\sigma^2 + \alpha^2} + \frac{4\alpha^3}{\left(4\sigma^2 + \alpha^2\right)^2} \right)|_{\alpha=0,\beta=0} \tag{S57}$$

$$= \frac{\partial \Pi}{\partial \mu_z} \left( -\frac{4}{4\sigma^2 + \alpha^2} + \frac{8\alpha}{\left(4\sigma^2 + \alpha^2\right)^2} + \frac{12\alpha^2}{\left(4\sigma^2 + \alpha^2\right)^2} - \frac{16\alpha^4}{\left(4\sigma^2 + \alpha^2\right)^3} \right) |_{\alpha=0,\beta=0} \tag{S58}$$

$$= -\frac{4}{4\sigma^2} \frac{\partial \Pi}{\partial \mu_z} |_{\alpha=0,\beta=0} \tag{S59}$$

$$= -\frac{4}{4\sigma^2} \frac{1}{4} \sqrt{\frac{2}{\mu_z}} \mathcal{N} \left( \sqrt{\frac{\mu_z}{2}} \right) |_{\alpha=0,\beta=0} \tag{S60}$$

$$= -\frac{1}{4\sigma^2} \frac{2\sigma}{\gamma} \mathcal{N} \left( \frac{\gamma}{2\sigma} \right) \tag{S61}$$

$$= -\frac{1}{2\sigma\gamma} \mathcal{N} \left( \frac{\gamma}{2\sigma} \right) \tag{S62}$$

Therefore, from *Equation S54* we obtain:

$$\frac{\partial^4 \mathcal{L}}{\partial \alpha^2 \beta^2} |_{\alpha=0,\beta=0} = -\frac{\gamma}{4\sigma^3} \mathcal{N} \left( \frac{\gamma}{2\sigma} \right) \tag{S63}$$

We therefore find that the magnitude of this term becomes smaller with increasing noise. This is why we observed larger values of $\beta = \frac{\Delta \mathbf{r}_{\text{rel}}^\top \Delta \mathbf{r}_{\text{irrel}}}{\|\Delta \mathbf{r}_{\text{rel}}\| \|\Delta \mathbf{r}_{\text{irrel}}\|}$ (*Equation S20*) in *Figure 4g* with increasing $\sigma$.

