## [Editor Report · eLife assessment]

This work provides a **valuable** analysis of the effect of two commonly used hyperparameters, noise amplitude and firing rate regularization, on the representations of relevant and irrelevant stimuli in trained recurrent neural networks (RNNs). The results suggest an interesting interpretation of prefrontal cortex (PFC) dynamics, based on comparisons to previously published data from the same lab, in terms of decreasing metabolic cost during learning. The evidence indicating that the mechanisms identified in the RNNs are the same ones operating in PFC was considered **incomplete**, but could potentially be bolstered by additional analyses and appropriate revisions.

---

## [Referee Report · Reviewer #1 (Public Review)]

Summary:

This study compares experimental data recorded from the PFC of monkeys to the activity of recurrent neural networks trained to perform the same `task' as the monkeys, namely, to predict the delivery of reward following the presentation of visual stimuli. The visual information varied along 3 dimensions, color, shape, and width. Shape was always relevant for reward prediction, width was always irrelevant, and color was irrelevant at the beginning of the trial but became relevant later on, once it could be assessed together with shape. The neural data showed systematic changes in the representations of these features and of the expected reward as the learning progressed, and the objective of this study was to try to understand what principles could underlie these changes. The simulations and theoretical calculations indicated that the changes in PFC activity (throughout learning and throughout a trial) can be understood as an attempt by the circuitry to use an efficient representational strategy, i.e., one that uses as few spikes as possible, given that the resulting representation should be accurate enough for task performance.

Strengths:

- The paper is concise and clearly written.

- The paper shows that, in a neural circuit, the information that is decodable and the information that is task-relevant may relate in very different ways. Decodable information may be very relevant or very irrelevant. This fact is critical for interpreting the results of pure decoding studies, which often assume an equivalence. This take-home message is not emphasized by the authors, but I think is quite important.

- The results provide insight as to how neural representations may be transformed as a task is learned, which often results in subtle changes in selectivity and overall activity levels whose impact or reason is not entirely clear just by looking at the data.

Weaknesses:

The match between the real PFC and the model networks is highly qualitative, and as noted by the authors, comparisons only make sense in terms of *changes* between early and late learning. The time scales, activity levels, and decoding accuracies involved are all different between the model and recording data. This is not to disregard what the authors have done, but simply to point out an important limitation.

---

## [Referee Report · Reviewer #2 (Public Review)]

Summary:

The study investigates the representation of irrelevant stimuli in neural circuits using neural recordings from the primate prefrontal cortex during a passive object association task. They find a significant decrease in the linear decodability of irrelevant stimuli over the course of learning (in the time window in which the stimuli are irrelevant). They then compare these trends to RNNs trained with varying levels of noise and firing rate regularization and find agreement when these levels are at an intermediate value. In a complementary analysis, they found (in both RNNs and PFC) that the magnitude of relevant and irrelevant stimuli increased and decreased, respectively, during learning. These findings were interpreted in terms of a minimization of metabolic cost in the cortex.

To understand how stimuli can be dynamically suppressed at times when they are irrelevant, the authors constructed and analyzed a reduced two-neuron model of the task. They found a mechanism in which firing rate regularization increased the probability of negative weights in the input, pushing the neural activities below the threshold. A similar mechanism was observed in RNNs.

Strengths:

The article is well-written and the figures are easily understood. The analyses are well explained and motivated. The article provides a valuable analysis of the effect of two parameters on representations of irrelevant stimuli in trained RNNs.

Weaknesses:

(1) The mechanism for suppressing dynamically relevant stimuli appears to be incomplete and does not explain clearly enough how representations of 'color' which are suppressed through negative input weights become un-suppressed in the presence of the second variable 'shape'.

(2) Interpretation of results in terms of the effect of metabolic cost on cortical dynamics is not backed up by the presented data/analyses. The change in dynamics of 'color' representations in the prefrontal cortex only qualitatively matches RNN dynamics and may arise from other causes.

---

## [Referee Report · Reviewer #3 (Public Review)]

Summary:

In order to study the factors and neural dynamics that lead to the suppression of irrelevant information in the brain, the authors trained artificial neural networks in the execution of a task that involved the discrimination of complex stimuli with three main features: color, shape, and width. Specific combinations of color and shape led to a reward, but the temporal structure made color dynamically irrelevant at the beginning of the trial, and then it became relevant once the shape was presented. On the other hand, the width of the stimulus was always irrelevant. Importantly, non-human primates were also trained to execute this task (in a previous study by the authors) and the activity from neural populations from the dorsolateral Prefrontal Cortex (dlPFC) was recorded, allowing to compare the coding of information by the artificial neural network model with what happens in biological neural populations.

The authors changed systematically the amount of noise present in the neural network model, as well as limiting the firing rate of the artificial neurons to simulate the limitations imposed by high metabolic costs in biological neurons. They found that models with medium and low noise, as well as medium and low metabolic cost, developed information encoding patterns that resembled the patterns observed throughout learning in the dlPFC, as follows: early in the learning process, color information was strongly represented during the whole trial, as well as shape and width, whereas the color/shape combination significance (XOR operation) was weakly encoded. Late in learning, color information was initially suppressed (while it was deemed irrelevant) and became more prominent during the shape presentation. Width information coding decreased, and the XOR operation result became more strongly encoded.

Subthreshold activity dynamics were studied by training artificial networks consisting of 2 neurons, with the aim of understanding how dynamically irrelevant information is suppressed and then encoded more strongly at a different time during the trial. Under medium noise and medium metabolic cost, color information is suppressed by the divergence of the activity away from the level that triggers spikes. The authors claim that this subthreshold dynamic explains the suppression of irrelevant information in biological neural networks.

Strengths:

The study leverages the power of computational models to simulate biological networks and do manipulations that are difficult (if not impossible) to perform in vivo. The analyses of the activity of the network model are neat and thorough and provide a clear demonstration of how noise and metabolic costs may affect the information coding in the brain. The mathematical analyses are rigorous and nicely documented.

Weaknesses:

The study does not leverage the fact that they have access to the activity of individual neurons both on a neural network model and in neural recordings. The model/brain comparison results are limited to the decodability of different pieces of information during the execution of the task at different stages of learning. It would have been useful if the authors had shown response profiles of individual neurons, both biological and artificial, to strengthen the claim that the activity patterns are similar. Perhaps showing that the firing rates vary in a similar way in the large models (like they do for the 2-neuron model) would have been informative. For instance, it is possible that suppression is not occurring in the dlPFC, but that the PFC receives input with this information already suppressed. If suppression indeed happens in the PFC, response profiles associated with this process may be observed.

There is no way to say that the 2-neuron models are in any way informative of what happens in brain neurons, or even larger artificial networks since the sources of sensory input, noise, and inhibition will differ between biological and artificial networks. And because the firing patterns are not shown for large networks, it is not clear if some non-coding artificial neurons will become broadly inhibitory but maintain a relatively high firing rate (to mention only one possibility).

---

## [Author Response]

**Reviewer 1:**

Many thanks for your positive review and clear overview of our paper. We also agree with your interpretation of our results that ‘the information that is decodable and the information that is task-relevant may relate in very different ways’ and we could have emphasised this point more in the paper.

With regards to the qualitative similarities between our models and our data, we agree that due to the fact that one can achieve any desired level of activity, decoding accuracy, performance, etc in a model, we focussed on *changes* over learning of key metrics that are commonly used in the field. Although this can appear qualitative at times because the raw values can differ between the data and our models, our main results are ultimately strongly quantitative (e.g., Fig. 3c,d, Fig. 4h,i, and Fig. 5f). We note that we could have fine tuned the models to have similar activity levels, decoding accuracies etc to our data, and on the face of it this may have made the results appear more convincing, but we felt that such trivial fine tuning does not change any of our key results in any fundamental way and is not the aim of computational modelling. The model one chooses to analyse will always be abstracted from biology in some way, by definition.

**Reviewer 2:**

Thank you very much for your kind comments and clear overview of our paper. We also hope that our paper ‘provides a valuable analysis of the effect of two parameters on representations of irrelevant stimuli in trained RNNs.’

With regards to our suggested mechanism of suppressing dynamically irrelevant stimuli, we are sorry that we did not provide a sufficient enough explanation of suppressing color representations when they are irrelevant. We hopefully provide a longer explanation here. Our mechanism of suppression of dynamically irrelevant stimuli does not suggest that it becomes un-suppressed later, only the behaviourally relevant variable should be decodable when it is needed (i.e., XOR). Although color decodability did increase slightly in the data and some of the models from the color period to the shape period, it was typically not significant and was therefore not a result that we emphasise in the paper (although this could be analysed further to see if additional mechanisms might explain it). We emphasise throughout that color decoding is typically similar between color and shape periods (either high or low) and either decreases or increases over time in both periods. We also focus on whether color decodability increases or decreases over learning during the color period when it is irrelevant (which we call ‘early color decoding’). Importantly, decoding of color or shape is not needed to perform the task, only decoding of XOR is needed to perform the task. For example, in our two-neuron networks, we observe perfect XOR decoding and only 75% decoding of color and shape, and decoding during the shape period is the same as the network at initialisation before any training. The mechanism we suggest of suppressing dynamically irrelevant stimuli does not predict that that stimulus should be un-suppressed later, only the behaviourally relevant variable should be decodable (i.e., XOR). Instead, what we try to explain is that color inputs can generate zero firing rate during the color period, when that input does not need to be used and is therefore irrelevant (and color decoding decreases during the color period over learning), but these inputs can be combined with shape inputs later to create a perfectly decodable XOR response.

With regards to interpretation of our results based on metabolic cost constraints, we feel that this is an unnecessarily strong criticism to say that it ‘is not backed up by the presented data/analyses.’ All of our models were trained with only a metabolic cost constraint, a noise strength, and a task performance term. Therefore, the results of the models are directly attributable to the strength of metabolic cost that we use. Additionally, although one could in principle pick any of infinitely many different parameters to change and measure the response in an optimized network, varying metabolic cost and noise are two of the most fundamental phenomena that neural circuits must contend with, and many studies have analysed the impact they have on neural circuit dynamics. Furthermore, in line with previous studies (Yang et al., 2019, Whittington et al., 2022, Sussillo et al., 2015, Orhan et al., 2019, Kao et al., 2021, Cueva et al., 2020, Driscoll et al., 2022, Song et al., 2016, Masse et al., 2019, Schimel et al., 2023), we operationalized metabolic cost in our models through L2 firing rate regularization. This cost penalizes high overall firing rates. (Such an operationalization of metabolic cost also makes sense for our models because network performance is based on firing rates rather than subthreshold activities.) There are however alternative conceivable ways to operationalize a metabolic cost; for example L1 firing rate regularization has been used previously when optimizing neural networks and promotes more sparse neural firing. Interestingly, although our L2 is generally conceived to be weaker than L1 regularization, we still found that it encouraged the network to use purely sub-threshold activity in our task. The regularization of synaptic weights may also be biologically relevant because synaptic transmission uses the most energy in the brain compared to other processes (Faria-Pereira et al., 2022, Harris et al., 2012). Additionally, even subthreshold activity could be regularized as it also consumes energy (although orders of magnitude less than spiking (Zhu et al., 2019)). Therefore, future work will be needed to examine how different metabolic costs affect the dynamics of task-optimized networks. Finally, with regards to our data analysis, we show that firing rates indeed decrease significantly over the course of learning (Fig. 5f), which is a direct prediction of neural circuits contending with metabolic cost constraints and is reflected in our networks trained under an L2 firing rate metabolic cost.

With regards to color representations in PFC only qualitatively matching those in our models, in line with the comment from Reviewer 1, we agree that due to the fact that one can achieve any desired level of activity, decoding accuracy, performance, etc in a model, we focussed on changes over learning of key metrics that are commonly used in the field. Although this can appear qualitative at times because the raw values can differ between the data and our models, our main results are ultimately strongly quantitative (e.g., Fig. 3c,d, Fig. 4h,i, and Fig. 5f). We note that we could have fine tuned the models to have similar activity levels, decoding accuracies etc to our data, and on the face of it this may have made the results appear more convincing, but we felt that such trivial fine tuning does not change any of our key results in any fundamental way and is not the aim of computational modelling. The model one chooses to analyse will always be abstracted from biology in some way, by definition. Finally, of course we note that changes in color decoding could result from other causes, but we focussed on two key phenomena that neural circuits must contend with: noise and metabolic costs. Therefore, it is likely that these two variables play a strong role in stimulus representations in neural circuits

**Reviewer 3:**

Thank you very much for your thorough and clear overview of our paper and we agree that it is important to investigate phenomena and manipulations in computational models that are almost impossible to do in vivo and we are pleased you found our mathematical analyses rigorous and nicely documented.

Although we agree that it can be useful to study the responses of individual neurons, we focussed on population analyses of all available neurons without omitting or specifically selecting neurons based on their dynamics. We are also not suggesting that the activities of individual ‘neurons’ in the models and data should be similar since our models are highly abstract firing rate models. But rather, the overall computational strategy, which one can access through population decoding and cross-generalised decoding, was what we were interested in comparing between the models and the data and is arguably the correct level of analysis of such models (an data) given our key questions (Vyas et al., 2020, Churchland et al., 2012, Mante et al., 2013, Ebitz et al., 2021).

We also certainly agree and are more than open to the fact that suppression of irrelevant stimuli may already be happening on the inputs arriving in PFC. Indeed, we actually suggest this as the mechanism in Fig. 5 (together with recurrent circuit dynamics that make use of these inputs).

With regards to the dynamics of the two-neuron networks not being ‘informative of what happens in brain networks’, we agree that these models are very simplified and may only contain very fundamental similarities with biological neurons. However, we only used them to illustrate the fundamental mechanism of generating zero firing rate during the color epoch so that it is more easily understandable for readers as they can see the entire 2-dimensional state space and the entire computational strategy can be seen (Fig. 5a-d). We also note that we did this for both rectified linear and tanh networks, thus showing that such a mechanism is preserved across fundamentally different firing rate nonlinearities. Additionally, after illustrating this fundamental mechanism of networks receiving color information but generating zero firing rate, we show that the exact same mechanism is at play in the large networks we use throughout the paper (Fig. 5e). We also only compare the large networks to our neural recordings. We do agree though that it would be interesting to further compare fundamental similarities and differences between our models and our neural recordings (always at the right level of analysis that makes sense for our chosen models) to show that the mechanisms we uncover in our models are also strongly relevant for our data.